# Characterizing cell-type spatial relationships across length scales in spatially resolved omics data

Rafael dos Santos Peixoto [1,2], Brendan F. Miller[1,2], Maigan A. Brusko[3], Gohta Aihara[1,2], Lyla Atta [1,2], Manjari Anant[1,4], Mark A. Atkinson[3], Todd M. Brusko [3], Clive H. Wasserfall[3] & Jean Fan [1,2] ✉

Spatially resolved omics (SRO) technologies enable the identification of cell types while preserving their organization within tissues. Application of such technologies offers the opportunity to delineate cell-type spatial relationships, particularly across different length scales, and enhance our understanding of tissue organization and function. To quantify such multi-scale cell-type spatial relationships, we present CRAWDAD, Cell-type Relationship Analysis Workflow Done Across Distances, as an open-source R package. To demonstrate the utility of such multi-scale characterization, recapitulate expected cell-type spatial relationships, and evaluate against other cell-type spatial analyses, we apply CRAWDAD to various simulated and real SRO datasets of diverse tissues assayed by diverse SRO technologies. We further demonstrate how such multi-scale characterization enabled by CRAWDAD can be used to compare cell-type spatial relationships across multiple samples. Finally, we apply CRAWDAD to SRO datasets of the human spleen to identify consistent as well as patient and sample-specific cell-type spatial relationships. In general, we anticipate such multi-scale analysis of SRO data enabled by CRAWDAD will provide useful quantitative metrics to facilitate the identification, characterization, and comparison of cell-type spatial relationships across axes of interest.

Spatially resolved omics (SRO) technologies enable molecular profiling to facilitate the identification of distinct cell types while preserving their spatial organization within tissues, providing an opportunity to evaluate cell-type spatial relationships. Cell-type spatial relationships such as colocalizations, defined as which cell types are spatially near each other, and separations, defined as which cell types are spatially away from each other, may exhibit distinct trends relevant to healthy tissue function[1] as well as disease[2]. As such, evaluating these cell-type spatial relationships provide an opportunity to advance our understanding of the association between cell-type organization, tissue function, and disease.

Cell-type spatial relationships can occur at different length scales, with some cell types colocalizing to engage in paracrine signaling and other close-range interactions at a fine, micrometer length scale[3]; others colocalizing into distinct environments and functional tissue units at a more meso-scale[1]; while others colocalizing into anatomical structures at a more macro-scale (Fig. 1a, b). Whether we consider two cell types as being colocalized is often a function of the spatial extent that we analyze (Fig. 1b). For example, two cell types uniquely present in distinct layers of the brain may be considered separated if we analyze only the spatial extent of the brain. However, we may consider these cell types to be colocalized

[1]Center for Computational Biology, Whiting School of Engineering, Johns Hopkins University, Baltimore, MD, USA. [2]Department of Biomedical Engineering, Johns Hopkins University, Baltimore, MD, USA. [3]Department of Pathology, Immunology, and Laboratory Medicine, University of Florida, Gainesville, FL, USA. [4]Department of Neuroscience, Johns Hopkins University, Baltimore, MD, USA. ✉e-mail: jeanfan@jhu.edu

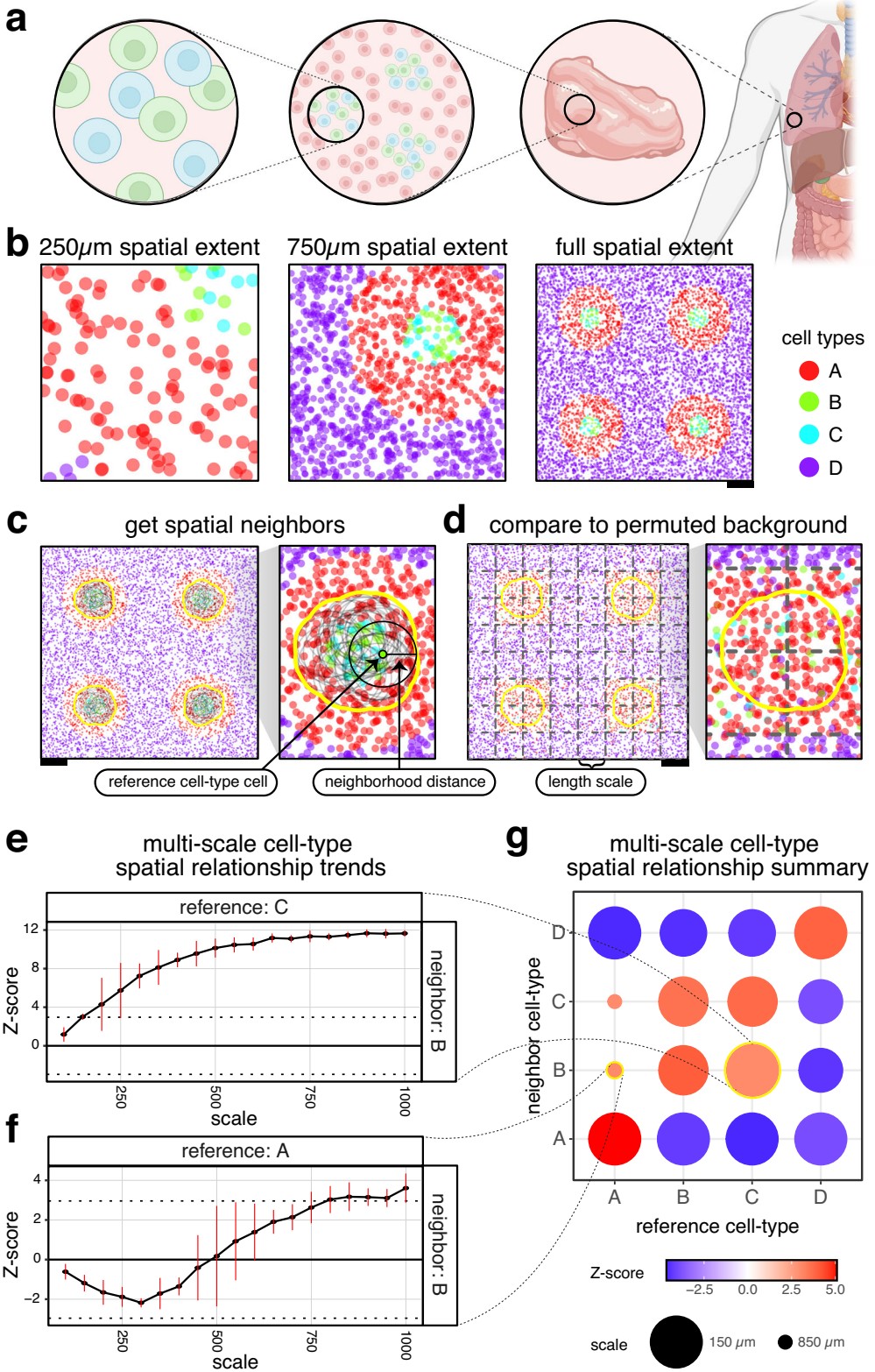

in the same organ if we analyze the spatial extent of the whole body. Thus, we sought to consider the effects of spatial extent by investigating cell-type spatial relationships across different length scales.

To quantitatively evaluate such cell-type spatial relationships across length scales, we present Cell-type Relationship Analysis Workflow Done Across Distances (CRAWDAD). We demonstrate CRAWDAD on simulated as well as real SRO datasets for diverse tissues assayed by Slide-seqV2[4], seqFISH[5], Xenium[6], MERFISH[7], and CODEX[8], though CRAWDAD is amenable to any SRO data for which cell positions and cell-type annotations can be obtained. CRAWDAD is available as an open-source R package at https://github.com/JEFworks-Lab/CRAWDAD with additional documentation and tutorials available at https://jef.works/CRAWDAD/.

**Fig. 1 | Motivating Cell-type Relationship Analysis Workflow Done Across Distances (CRAWDAD) using simulated data. a** Illustration of the cell-type spatial relationships found at different length scales. **b** Simulated spatial omics tissue data, visualized at different scales. Each point is a cell, colored by cell type. Scale bars correspond to 250 μm. **c** Representation of the creation of the neighborhood and the null background. CRAWDAD draws a circle (neighborhood distance as the radius) around each cell of the reference cell type and merges them into one neighborhood. **d** CRAWDAD creates a grid of side-by-side tiles (length scale defined as the side length for square tiles and the distance between opposite edges for hexagonal tiles) and shuffles the labels inside each tile to create the null background. **e, f** The multi-scale spatial relationship trend plots for **e.** reference cell-type C and neighbor cell-type B and **f.** reference cell-type A and neighbor cell-type B. The horizontal black dotted lines represent the Z-score significance threshold corrected for multiple testing (Z-score = ±2.96). The vertical red bars represent the error bars of ± one standard deviation from the mean Z-score estimated using permutations. **g** Summary visualization of all cell-type spatial relationships. The size of the dot represents the scale in which a neighbor cell type first reaches a significant spatial relationship with respect to a reference cell type. The color of the dot is the Z-score at such scale. Created in BioRender. Fan, J. (2023) BioRender.com/y47n964.

## Results

### Overview of method

Given cell centroid positions and their cell-type annotations, CRAWDAD evaluates the statistical enrichment or depletion of each cell type within the spatial neighborhood around cells of a reference cell type at a particular spatial length scale. To achieve this, CRAWDAD first draws a neighborhood around cells of a reference cell type based on a user-defined neighborhood distance d and calculates the proportion of every cell type inside this neighborhood, excluding the original reference cell that seeded it (Fig. 1c). Next, CRAWDAD creates a series of non-overlapping grids of square or hexagonal tiles where the size of each tile corresponds to a user-defined spatial length scale. Then, it shuffles the cell-type annotations for all cells within each tile to create an empirical null background at the specified spatial length scale (Fig. 1d). Lastly, given the observed and shuffled cell-type annotations, CRAWDAD uses a binomial proportion testing framework to evaluate if the observed cell-type proportions are significantly different from what is expected by chance based on the shuffled data. In this process, we obtain a Z-score for each neighbor and reference cell-type pair at the given spatial length scale. To assess statistical significance, we use default Z-scores of ±1.96 (p-value = 0.05) with multiple testing correction based on the number of cell-type combinations evaluated. For a given cell-type pair, if their Z-score is above the positive significance threshold, we interpret the neighbor cell type as being enriched in the neighborhood of the reference cell type; if their Z-score is below the negative significance threshold, we interpret the neighbor cell type as being depleted in the neighborhood of the reference cell type. In addition, we define colocalization as the mutual enrichment of cell types within each other's neighborhoods. Alternatively, we define separation as the mutual depletion of the cell types within each other's neighborhoods.

To evaluate spatial relationships with different spatial extents, CRAWDAD repeats this process for a series of spatial length scales. In this manner, we obtain a Z-score for each cell-type pair in each spatial length scale. We can visualize the results by plotting the Z-scores on the yaxis and the length scales on the x-axis, creating a multi-scale spatial relationship trend plot for each cell-type pair. Here, we refer to the scale of the relationship between two cell types as the first evaluated spatial length scale in which the Z-score is above the significance threshold. To summarize spatial relationships across all cell-type pairs, we plot the scale of the relationships and associated Z-scores as a dot plot. To mitigate potential grid edge effects, CRAWDAD uses different random seeds and grid offsets to create multiple permutations of the shuffled data. The Z-scores obtained for each cell-type pair at each length scale are then averaged across permutations.

### CRAWDAD characterizes cell-type spatial relationships at multiple length scales in simulated data

To highlight the utility of evaluating cell-type spatial relationships across multiple length scales, we first simulated 8000 cells representing four cell types in a 2000 μm-by-2000 μm tissue sample (Methods). In this simulated dataset, cell-types B and C are spatially intermixed with one another, while cell-type A forms a distinct structure around them that further isolate cell-types B and C from cell-type D (Fig. 1b). We applied CRAWDAD to characterize the spatial relationships between each cell-type pair across length scales ranging from 100 μm to 1000 μm using a neighborhood size of 50 μm (Methods). We found that the neighborhood around reference cell-type C is significantly enriched with cells of cell-type B from a fine length scale of ~150 μm (Fig. 1e, g). Likewise, we observed that the neighborhood around reference cell-type B is significantly enriched with cells of cell-type C from a similarly fine length scale (Supp. Fig. 1a). Given this mutual enrichment relationship, we interpret cell-types B and C to be spatially colocalized from this fine length scale. In contrast, we noted cell-types A and B to be spatially colocalized from a comparably coarser length scale of ~800 μm (Fig. 1f, g, Supp. Fig. 1b).

We note that some multi-scale cell-type spatial relationship trends will exhibit a monotonic behavior, tending towards only enrichment or depletion, while others might oscillate between enrichment and depletion. For example, the neighborhood around reference cell-type C exhibits an increasing Z-score with respect to neighbor cell-type B across length scales, resulting in a monotonically increasing spatial trend towards enrichment (Fig. 1e). In contrast, the neighborhood around reference cell-type A exhibits an initially decreasing Z-score with respect to neighbor cell-type B that eventually becomes a positive Z-score at larger length scales, resulting in an oscillatory spatial trend (Fig. 1f). Such an oscillatory spatial trend better reflects how cell-types A and B are separated in different compartments but ultimately comprise the same structures evident only at larger spatial extents. Other methods that evaluate spatial relationships only at the whole tissue scale may miss these distinctions regarding monotonic and oscillatory spatial trends. For example, Squidpy's neighborhood enrichment implementation of the approach described by Schapiro et al.[9] calculates an enrichment score based on the proximity of cell types from a spatial neighborhood graph created using a fixed distance[10]. Therefore, it does not capture the dynamics of such multi-scale cell-type spatial relationships (Supp. Fig. 1c).

CRAWDAD's quantified cell-type spatial relationships are also robust as the spatial extent is broadened to a larger tissue section inclusive of new cell types. To demonstrate this, we expanded the tissue in the simulated data to include another cell type, cell-type E, not present in the original tissue (Supp. Fig. 1d). When we apply CRAWDAD to this expanded tissue, the spatial relationships for cell-types A, B, and C in terms of the quantified Z-scores remain the same (Fig. 1e, Supp. Fig. 1f). In this manner, the spatial relationships for cell-types A, B, and C are robust to the addition of the new cell type in expanded regions not considered in the original analysis. In contrast, when we apply Squidpy's neighborhood enrichment implementation to the original versus the expanded tissue, the neighborhood enrichment metrics for cell-types A, B, and C are altered (Supp. Fig. 1c, e).

To further benchmark and compare CRAWDAD's functionality, we simulated a variety of SRO datasets using a previously developed simulation framework[11] (Methods). Briefly, we simulated cells by sampling from a uniform distribution to create x-y positions. We split the cells into two groups, and, for each group, we associated a value to each cell using independent, autocorrelated Gaussian random fields

(Supp. Fig. 2a). We binarized the values, splitting the cells into two cell types based on the underlying simulated value (Supp. Fig. 2b). In this manner, we created a simulated dataset with four cell types (Supp. Fig. 2c) where we expect each cell type to be enriched with itself due our use of spatially autocorrelated simulation values. Likewise, we can expect the two cell types simulated from the same Gaussian random field to be spatially mutually exclusive and therefore identified to be separated. Additionally, we expect the cell types from different random fields to exhibit no significant spatial relationship due to the Gaussian random fields being independent. We repeated this process to create a total of ten random simulated datasets.

We used these simulated datasets to benchmark and compare CRAWDAD with two other spatial relationship analysis methods that also consider spatial length scales, Ripley's K Cross[12] and Squidpy's co-occurrence implementation of the approach described in Tosti et al.[13]. Although all evaluated methods perform cell-type enrichment analysis across length scales, their definition of length scales differs. Briefly, Ripley's K Cross evaluates multiple length scales by increasing the neighborhood size while comparing the cell-type proportion in the neighborhood to the global proportion. On the other hand, Squidpy's co-occurrence implementation evaluates multiple length scales by increasing the size of an annulus neighborhood and calculating the conditional probability of the neighbor cell types given the reference cell type. In addition, as Ripley's K Cross and Squidpy's co-occurrence implementation do not present a threshold to determine statistical significance, for comparative purposes, we opted to assess each method's ability to distinguish between cell-type spatial enrichment and depletion. Specifically, given a reference cell type, we considered a method as achieving a true positive prediction if the cell type identified with the most enriched relationship trend was itself. Alternatively, we also considered a method as achieving a true positive prediction if the cell type identified with the most depleted relationship trend was the other cell type from the same Gaussian random field. We identified the cell type with the most enriched and most depleted spatial trend using their area under the trend curve value (Methods). Using this approach, we evaluated all four cell types across all ten simulated datasets using all three methods (Supp. Fig. 2d). Based on this simulation framework, we obtained a true positive rate of 0.95 for CRAWDAD, 0.86 for Squidpy's co-occurrence implementation, and 0.8 for Ripley's K Cross. In this manner, cell-type spatial relationships identified by CRAWDAD can more accurately distinguish between cell-type spatial enrichment and depletion compared to other evaluated methods.

**CRAWDAD recapitulates expected cell-type spatial relationships in the real spatial omics datasets of tissues**

We next applied CRAWDAD to evaluate cell-type spatial relationships in SRO datasets of real tissues. First, we analyzed a spatial transcriptomics dataset obtained from a highly spatially organized tissue, the mouse cerebellum, assayed by Slide-seqV2[4] and previously annotated by RCTD[14] (Fig. 2a). Because Slide-seqV2 uses 10 μm barcoded beads to profile the gene expression within tissues in a spatially resolved manner, spatially resolved measurements may not necessarily correspond to single cells. However, given that a typical animal cell is also roughly 10–20 μm in size[15], we assumed here that the observed spatial position and cell-type assignments associated with each bead generally reflects the spatial position and cell-type annotations of the cell within the immediate vicinity of that bead. As such, we treat Slide-seqV2 beads with non-doublet RCTD annotations as effectively single cells for the CRAWDAD analysis. We applied CRAWDAD to evaluate 10,098 annotated cells representing 19 cell types using a neighborhood size of 50 μm across length scales ranging from 100 μm to 1000 μm (Fig. 2b, Methods). Among the significantly colocalized cell-type pairs identified were Purkinje neurons and Bergmann glia

(Fig. 2b, c, Supp. Fig. 3a), which are known to interact at close distances within the Purkinje cell layer of the cerebellum[16]. Likewise, the neighborhood around Purkinje neurons was significantly depleted of oligodendrocytes and vice versa (Fig. 2b, c, Supp. Fig. 3b). As such, we may interpret these cell types as being spatially separated, consistent with the known spatially distinct layer structure of the cerebellum. Additionally, we applied Ripley's K Cross (Fig. 2d) and Squidpy's co-occurrence implementation (Fig. 2e) to the same dataset. We find that these other methods do not as clearly distinguish these expected cell-type spatial relationships. Specifically, when analyzing cell-type spatial relationships with Purkinje neurons as the reference cell type, we note that CRAWDAD's Z-score trend for Bergmann glia increases as the length scale increases, crossing the upper significance threshold and defining an enrichment of Bergmann glia among the neighborhood of Purkinje neurons, as expected (Fig. 2c). Likewise, CRAWDAD's Z-score trend for oligodendrocytes decreases as the length scale increases, crossing the lower significance threshold and defining a depletion of oligodendrocytes among the neighborhood of Purkinje neurons, as expected (Fig. 2c). These two cell-type trends are further distinct from other cell types in the cerebellum. This clear separation between these two cell-type trends is not observed in the other evaluated spatial analysis methods (Fig. 2d, e).

Next, we analyzed a single-cell resolution spatial transcriptomics dataset of a whole developing embryo assayed by seqFISH[5] (Fig. 2f). We applied CRAWDAD to evaluate cell-type spatial relationships for 19,416 cells representing 22 cell types using a neighborhood size of 50 μm across length scales ranging from 100 μm to 1000 μm (Fig. 2g, Methods). Consistent with the original publication's observations, CRAWDAD identified significant spatial colocalization between the intermediate and lateral plate mesoderm cells and significant spatial separation between intermediate mesoderm and cardiomyocyte cells (Fig. 2g, h, Supp. Fig. 3c, d). Again, such differences between cell-type spatial relationships are difficult to discern using on other evaluated spatial analysis methods (Fig. 2i, j).

To further exemplify CRAWDAD's applicability to potentially less well-organized tissues such as cancer tissues, we applied it to a breast cancer dataset assayed by Xenium[6] (Fig. 3a). We applied CRAWDAD to evaluate 162,107 annotated cells representing 19 cell types using a neighborhood size of 100 μm across length scales ranging from 100 μm to 1000 μm (Fig. 3, Methods). CRAWDAD identified three groups of cell types based on their cell-type spatial relationships, corresponding to histologically distinct structures (Fig. 3b, c).

In general, we note that the cell-type spatial relationships identified in CRAWDAD are not always symmetric. Asymmetric results may be caused by two scenarios: location imbalance and density imbalance. In location imbalance, cells of one cell type may be close to only some cells of the other cell type, but not all. For example, the neighborhood of UBCs is enriched with granule cells (Supp. Fig. 4a, c). However, UBCs are rare and present in only a small proportion of the granule cells' neighborhood and therefore does not represent a significant relationship (Supp. Fig. 4b, d). In density imbalance, cells from one type are highly concentrated in one region, with a few dispersed across other parts of the tissue. Therefore, the sparse cells will contribute to the creation of the neighborhood as the reference cell type but will not significantly contribute to the proportions as the neighbor cell type, due to their small number. For example, a large part of the presomitic mesoderm's neighborhood is created by its sparse cells, which encapsulate spinal cord cells, creating a relationship of enrichment (Supp. Fig. 4e, g). On the other hand, most of the presomitic mesoderm cells are outside the spinal cord's neighborhood creating a relationship of depletion (Supp. Fig. 4f, h). Such asymmetric cell-type spatial relationships may reflect non-exclusive cell-type interactions. For example, immune cells may infiltrate a focal tumor such that the neighborhood of tumor cells will be enriched with immune cells, but the neighborhood of immune cells might not be enriched by tumor

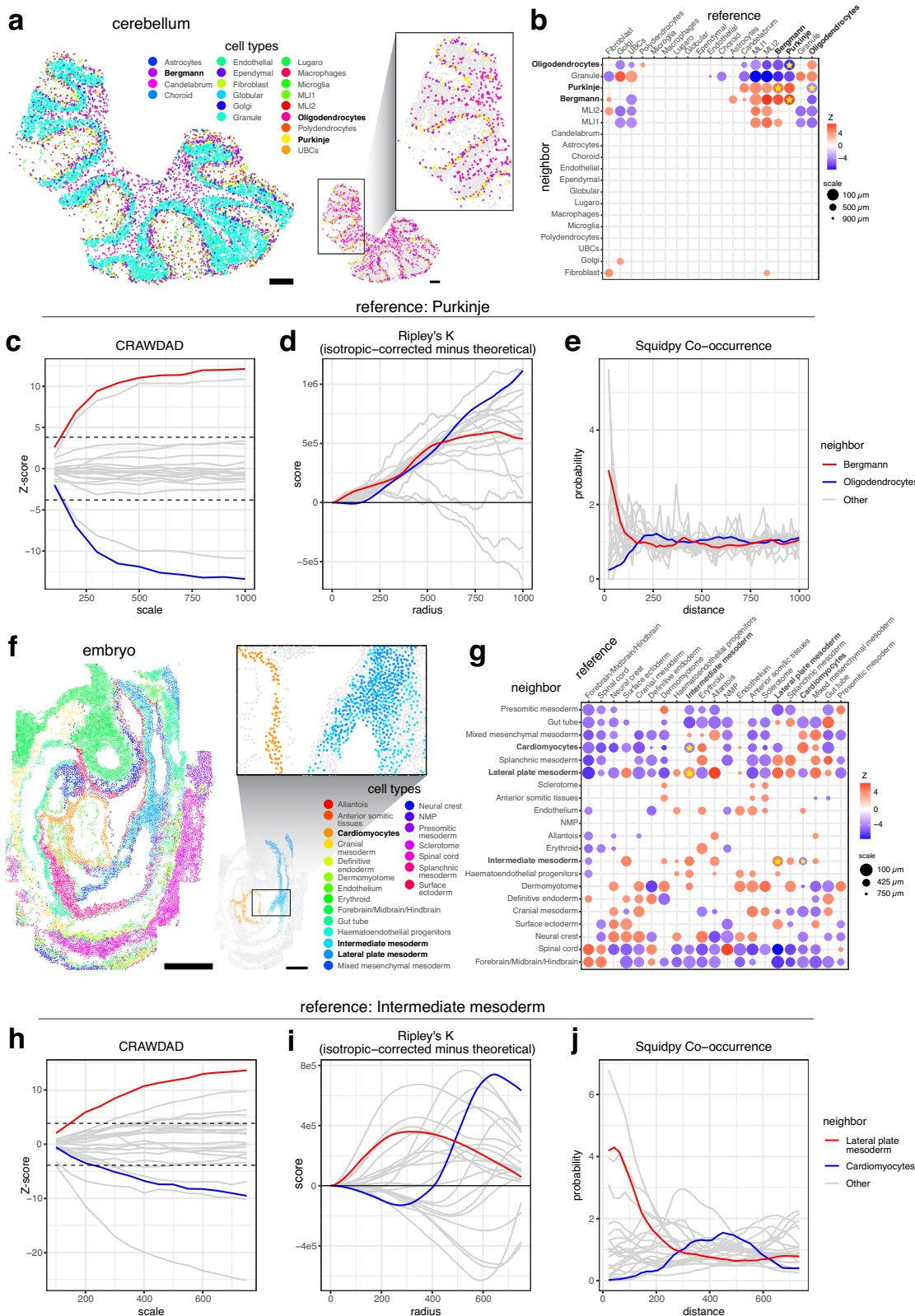

cells given their widespread spatial distribution throughout the body, consistent with a non-exclusive cell-type spatial relationship at a whole-body spatial extent[17]. Therefore, CRAWDAD can quantitatively capture such asymmetric cell-type spatial relationships and effectively delineate cell-type spatial relationships across multiple length scales for diverse tissues and SRO technologies.

## CRAWDAD enables comparison of cell-type spatial relationships across multiple samples

Beyond characterizing cell-type spatial relationships within a single sample, such multi-scale characterization enabled by CRAWDAD can also be used to compare cell-type spatial relationships across samples spanning different conditions, such as health and disease,

**Fig. 2 | CRAWDAD characterizes cell-type spatial relationships in the mouse cerebellum assayed by Slide-seqV2 and the mouse embryo assayed by seqFISH. a** Spatial visualization of cell-type annotations from RCTD in the cerebellum. Scale bars correspond to 250 μm. **b** Summary visualization all cell-type spatial relationships in the cerebellum data. Select cell types highlighted to correspond with (**c–e**). **c–e** The multi-scale spatial relationship trend plot for Purkinje neurons as the reference cell type for (**c**) CRAWDAD, (**d**) Ripley's K Cross, and (**e**) squidpy co-occurrence implementation of Tosti et al. with neighboring cell-types Bergmann glia and Oligodendrocytes highlighted in red and blue, respectively. All other neighboring cell types in gray. The horizontal black dotted lines in (**c**) represent the *Z*-score significance threshold corrected for multiple testing (Z-score = ±3.81). The vertical red bars in (**c**) represent the error bars of ± one standard deviation from the mean *Z*-score estimated using permutations. **f** Spatial visualization of annotated cell types in the embryo data. Scale bars correspond to 250 μm. **g** Summary visualization all cell-type spatial relationships in the embryo data. Select cell types highlighted to correspond with (**h–j**). **h–j** The multi-scale cell-type spatial relationship trend plot for Intermediate mesoderm cells as the reference cell type for (**h**) CRAWDAD, (**i**) Ripley's K Cross and (**j**) Squidpy co-occurrence implementation of Tosti et al. with neighboring cell-types Lateral plate mesoderm and Cardiomyocytes highlighted in red and blue, respectively. All other neighboring cell types in gray. The horizontal black dotted lines in (**h**) represent the *Z*-score significance threshold corrected for multiple testing (Z-score = ±3.88). The vertical red bars (**h**) represent the error bars of ± one standard deviation from the mean *Z*-score estimated using permutations.

development, or replicates. To demonstrate this functionality, we applied CRAWDAD to nine mouse brain samples assayed by MERFISH comprised of three replicates from three distinct Bregma locations[7] with cell-type annotations obtained previously using unified clustering[18] (Fig. 4a). We applied CRAWDAD to evaluate 734,693 annotated cells across all datasets representing 14 cell types using a neighborhood size of 50 μm across length scales ranging from 100 μm to 1000 μm (Methods). To compare multi-scale cell-type spatial relationships across samples, we calculated the signed area under the curve (AUC) for each Z-score trend for each cell-type pair. We then performed dimensionality reduction with principal component analysis (PCA) on all signed AUC values to find that cell-type spatial relationships of replicates from the same Bregma location are highly similar, as they are positioned closer together in PC space (Fig. 4b). We likewise overall observed a smaller variance in the signed AUC values within replicates from the same Bregma location compared to across locations (Fig. 4c). These results suggests that samples from the same Bregma location have cell-type spatial relationships that are more similar than those from different Bregma locations, as expected. Importantly, this similarity in cell-type spatial relationships is robust to tissue rotation and small local diffeomorphisms, as some of the brain tissue sections profiled are rotated with small tissue distortions compared to others. To further investigate specific highly variable cell-type spatial relationships, we visualized the spatial relationship trends for the cell-type pair with the highest signed AUC variance across locations: GABAergic Estrogen-Receptive Neurons as reference and Excitatory Neurons as neighbor (Fig. 4d). Despite its comparatively higher signed AUC variance across locations, samples from the same Bregma location still generally exhibited the same depletion trend whereas samples across Bregma locations varied (Fig. 4d). Visual inspection of GABAergic Estrogen-Receptive Neurons and Excitatory Neurons also suggested high consistency in terms of spatial relationships within replicates from the same Bregma location compared to across locations (Fig. 4e). As such, cell-type spatial relationship trends quantified by CRAWDAD can be used to contrast samples to confirm that cell-type spatial relationships in the mouse brain are generally highly consistent within replicates from the same Bregma location compared to across locations.

## CRAWDAD reveals functionally relevant cell-type spatial relationships in the human spleen

Finally, we sought to apply CRAWDAD to characterize and compare cell-type spatial relationships in SRO datasets of the human spleen. The spleen is a highly structured organ where cell types interact to filter blood and initiate immune responses. Delineating the spatial organization of cell types in the spleen can provide insights into how these different cellular populations may achieve such diverse immunologic functions. Therefore, we focused on six single-cell resolution spatial proteomics datasets of the human spleen comprised of two replicates from three individuals each assayed by CODEX[8] as part of The Human BioMolecular Atlas Program (HuBMAP)[19]. We first performed graph-

based clustering for one representative section of a human spleen with 154,446 cells to identify 12 cell types based on 28 protein markers (Fig. 5a–c, Methods). Applying CRAWDAD with a neighborhood of 50 μm and length scales ranging from 100 μm to 1750 μm, we observe cell-type spatial relationships to broadly recapitulate known splenic architecture. Cell types primarily colocalized at a coarse length scale into two major compartments defined by follicle B cells and red pulp B cells corresponding to the white pulp (WP) and red pulp (RP), respectively[20] (Fig. 5d). Consistent with the functional roles of these compartments, cell types identified to colocalize with red pulp B cells include macrophages, neutrophils, and monocytes that may remove old and dead red blood cells within the RP[21]. Likewise, cell types identified to colocalize with follicle B cells include CD4+ memory T cells and podoplanin-expressing cells within the WP (Fig. 5d). Podoplanin-expression has been previously associated with T-cell zone reticular cells[22] and observed to surround large arteries[23] within the WP.

To determine whether such cell-type spatial colocalization relationships are consistent both within and across individuals, we further repeated these analyses with 837,952 cells from five additional spleen samples. To ensure all datasets were annotated in a uniform manner, we applied batch correction[24] and used a linear discriminant analysis model to transfer cell-type annotations to these new datasets (Methods, Supp. Fig. 5a–c). We then applied CRAWDAD to identify similar cell-type spatial relationships corresponding to the WP and RP compartments both within and across individuals (Fig. 5d, Supp. Fig. 6a). Further analyzing the variance of the relationship trend's AUC values (Fig. 5e), we noticed that most cell-type spatial relationship trends were highly consistent across patients and samples, reflecting the ordered patterning of the functional tissue regions (Fig. 5e, f, Supp. Fig. 7a). Select cell-type spatial relationship trends had patient-specific relationships, exhibiting consistent trends within replicates from the same patient but varying across patients, suggestive of potential patient-specific variation (Fig. 5f, Supp. Fig. 7b). Other cell-type spatial relationship trends varied even within replicates, suggestive of potential tissue sample-specific patterns (Fig. 5f, Supp. Fig. 7c). In general, we anticipate assessing these variations in cell spatial relationships can give insight into inter- and intra-individual variation linked to donor and tissue-specific features.

Although our clustering analysis identified a cluster of cells highly expressing CD4 and CD45RO proteins, which we interpreted as CD4+ memory T cells, we were unable to distinguish follicular helper T cells, a specialized subset of CD4+ memory T cells that play a critical role in the adaptive immune response, due to limitations of our 28 protein-marker panel. However, we know that follicular helper T cells interact with B cells during the B cell maturation and differentiation process within the B cell follicles. As such, we hypothesized that follicular helper T cells would represent a spatially defined subset of these CD4+ memory T cells that are colocalized with follicular B cells[25]. To identify putative follicular helper T cells, we therefore applied the same binomial testing framework used in CRAWDAD's pairwise spatial relationship testing to identify CD4+ memory T cells that are statistically

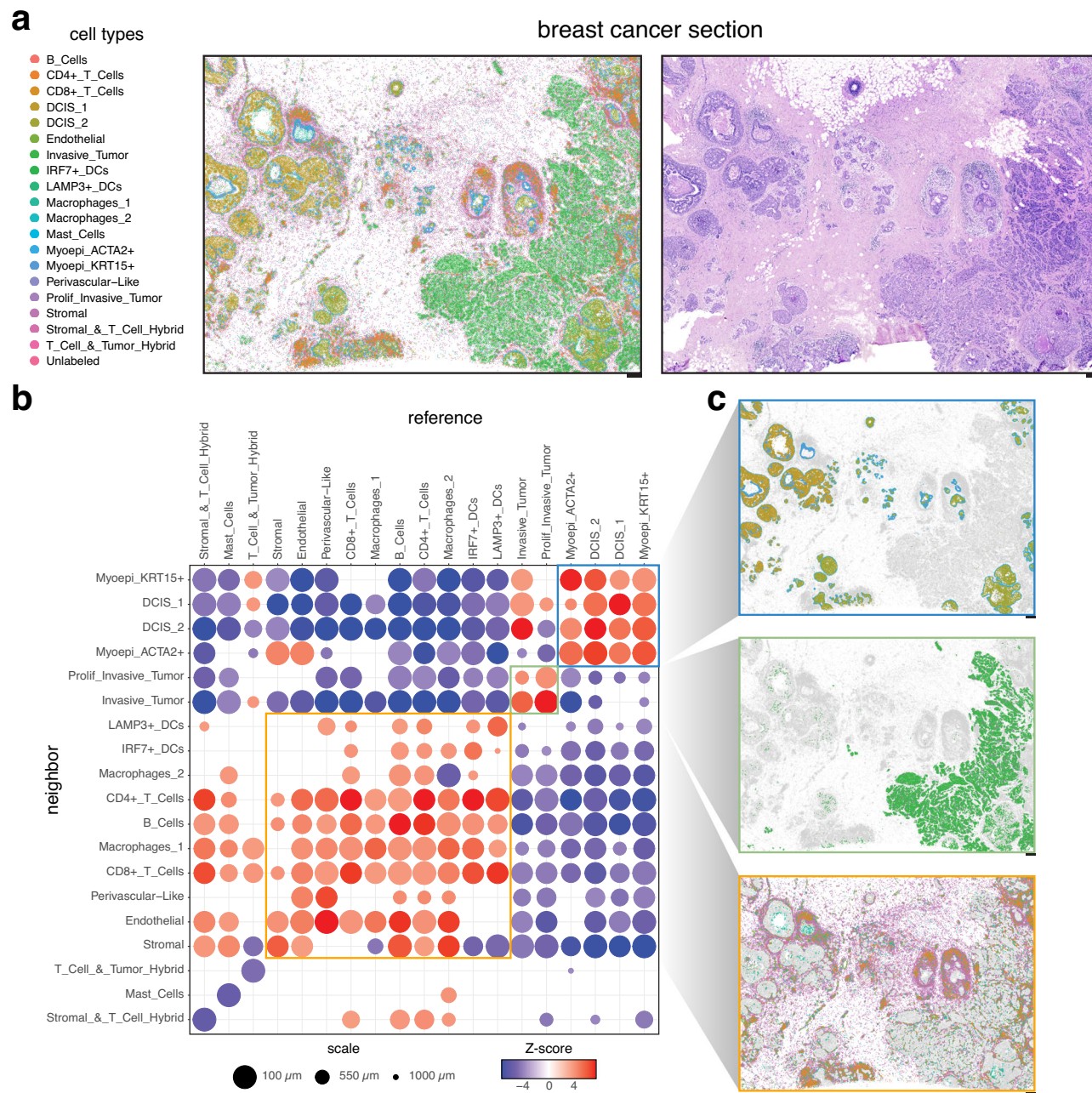

**Fig. 3 | CRAWDAD characterizes cell-type spatial relationships in breast cancer assayed by Xenium. a** Spatial visualization of annotated cell types (left) with corresponding histology image (right). **b** Summary visualization all cell-type spatial relationships in the breast cancer data. Select groups of consistently colocalized cell types are outlined by a unique color. **c** Spatial visualization of the consistently colocalized cell types, outlined by the corresponding group colors in (**b**). Scale bars correspond to 250 μm.

enriched around follicle B cells (Fig. 5g, Supp. Fig. 6b, Methods). Briefly, for each given cell, we perform an exact binomial test to verify if the proportion of each cell type inside the cell's neighborhood is significantly greater than the global proportion. We observed that the distribution of these putative follicular helper T cells as a percentage of all CD4+ memory T cells (37% to 52%, Fig. 5h) is consistent with previous characterizations within the spleen from healthy donors achieved through orthogonal fluorescence-activated cell sorting approaches[26,27].

## Discussion

In this paper, we present CRAWDAD, to quantify cell-type spatial relationships across spatial length scales. We validated CRAWDAD by

recapitulating expected cell-type spatial relationships in simulated and real SRO datasets of diverse tissues assayed by diverse SRO technologies. We demonstrated that our tool was able to provide distinct insights compared to existing spatial enrichment and multi-scale analyses. We emphasize that cell-type spatial relationships may vary across spatial scales and such multi-scale characterization enabled by tools like CRAWDAD can reveal distinct insights not apparent in evaluations of spatial relationships that consider only the whole tissue scale. Additionally, we emphasize that such quantified cell-type spatial relationships trends can be used to compare across SRO datasets and demonstrate its application in identifying consistent spatial trends within mouse brain replicates that are distinct across Bregma locations. We further apply CRAWDAD to characterize cell-type spatial

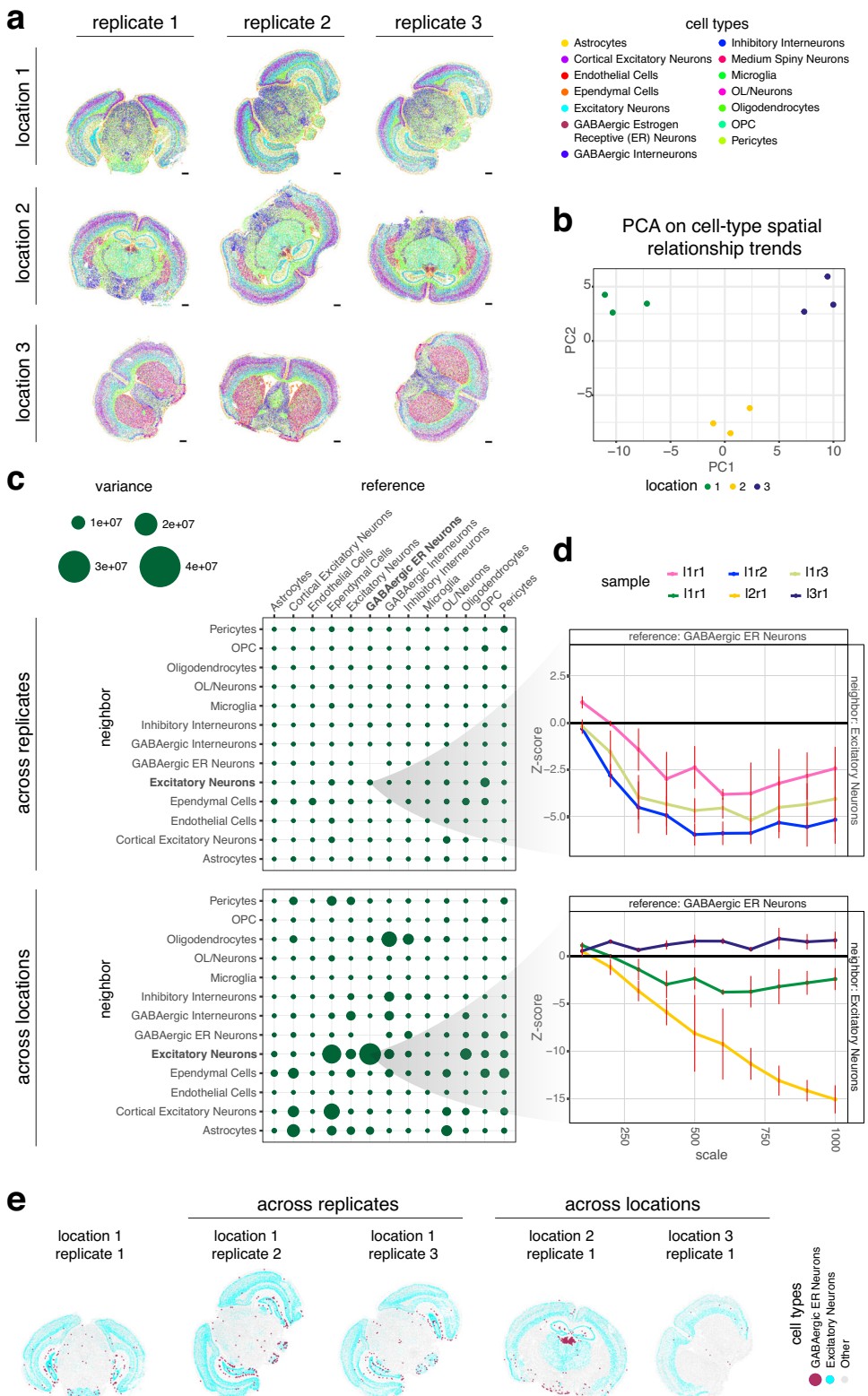

**Fig. 4 | CRAWDAD enables comparison of spatial relationships across different tissue sections of the mouse brain assayed by MERFISH. a** Spatial visualization of annotated cell types in each sample. Scale bars correspond to 1000 μm. **b** Visualization of each sample in the principal component space with the first two principal components calculated using the standardized signed AUC of each multi-scale cell-type spatial relationship trend. **c** Variability of multi-scale cell-type spatial relationship trends calculated as the variance of the AUC values across replicates from location 1 (top) and locations from replicate 1 (bottom). **d** The multi-scale cell-type spatial relationship trend plot of GABAergic Estrogen-Receptive (ER) Neurons as reference and Excitatory Neurons as neighbor for replicates from location 1 (top) and different locations from replicate 1 (bottom). The vertical red bars represent the error bars of ± one standard deviation from the mean Z-score estimated using permutations. **e** Spatial visualization of the GABAergic ER Neurons and Excitatory Neurons in replicates from location 1 (left) and different locations from replicate 1 (right).

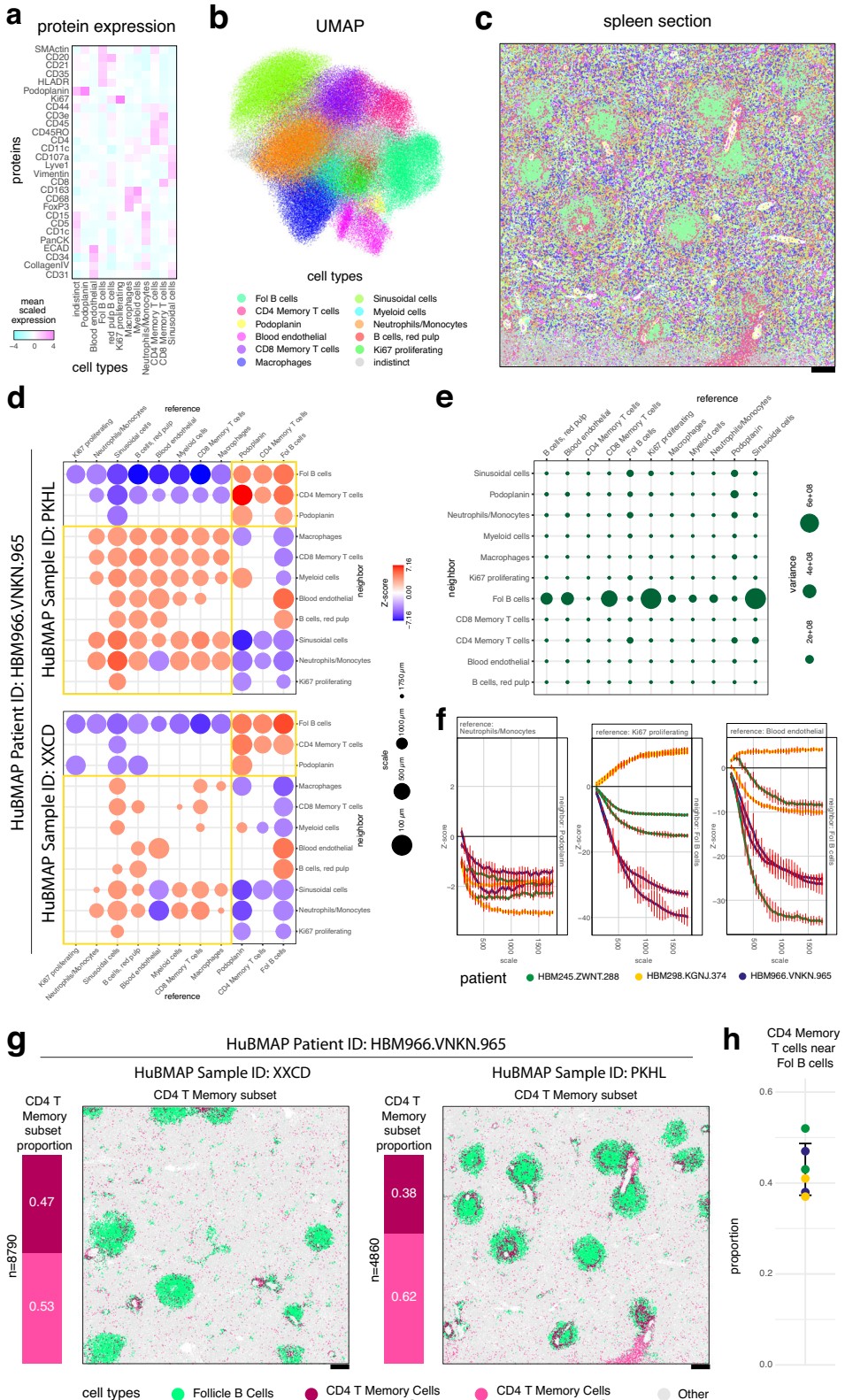

relationships to HuBMAP SRO datasets of the human spleen to identify generally consistent spatial trends reflective of the organization of the red and white pulp but also reproducible patient-specific variation, though the sample sizes evaluated here limit our ability to make general significant conclusions. As atlasing efforts such as HuBMAP[19], the Human Cell Atlas[28], and others continue to profile the spatial organization of cells within tissues, we anticipate identifying significant

spatial variation across axes of interest will become more feasible in the future, though additional scalable, comparative meta-analysis tools to integrate statistics from many samples across multiple studies in a manner that is robust to batch effects may be then needed. We expect that the incorporation of quantitative spatial trend metrics such as those provided by CRAWDAD will be useful in such meta-analyses to ultimately facilitate in the identification and

**Fig. 5 | CRAWDAD characterize cell-type spatial relationships in the human spleen assayed by CODEX. a–c** From the PKHL tissue section from patient HBM966.VNKN.965, **a** heatmap of marker protein expression for annotated cell types; **b** UMAP reduced-dimensional visualization of annotated cell types; **c** spatial visualization of annotated cell types in one representative tissue section. Scale bars correspond to 250 μm. **d** Summary visualization of the multi-scale cell-type spatial relationship analysis for tissue sections PKHL and XXCD from patient HBM966.VNKN.965. Cell types consistently colocalized in the white and red bulk are highlighted with small and large squares, respectively. **e** Variability of multi-scale cell-type spatial relationship trends calculated as the variance of the signed AUC values across samples. **f** The multi-scale cell-type spatial relationship trend plots are shown for select cell-type pairs exhibiting low variability across different samples, high variability across patients but low variability within replicates, and high variability across samples including within patients. The vertical red bars represent the error bars of ± one standard deviation from the mean Z-score estimated using permutations. **g** Subset of CD4+ Memory T cells near Follicle B cells. The number of CD4+ Memory T cells (*n*) and the proportion of subsets (left) and spatial visualization of subsets (right) in tissue sections PKHL and XXCD from patient HBM966.VNKN.965. **h** Proportion of CD4+ Memory T cells near Follicle B cells overall CD4+ Memory T cells in each sample. The black bars represent the error bars of ± one standard deviation from the mean proportion estimated using the samples.

characterization of cell-type spatial relationships in complex tissues to advance our understanding of the relationship between cell-type organization and tissue function.

Although we have demonstrated CRAWDAD to be a potentially useful tool in identifying, characterizing, and comparing cell-type spatial relationships, there are several considerations worth noting as they may influence interpretation. First, CRAWDAD results rely on a few user-defined parameters. In particular, it uses a fixed neighborhood distance *d* to determine the size of the neighborhood used to consider neighboring cells. In the context of geospatial analysis, such sensitivity of results to the neighborhood distance has been previously characterized as the sensitivity to kernel bandwidth[29]. We note that if the defined *d* is too small, the neighborhood will only contain cells from the reference cell type. In such a scenario, the total number of neighbor cells would be zero, leading to non-significant results. Alternatively, if *d* is too large, the neighborhood will encompass all the cells in the sample. In this case, the proportions of cell types within the neighborhood before and after shuffling will remain the same, leading to non-significant results. Generally, we recommend choosing *d* based on the biological constraints of the analysis. For example, to identify cell-type spatial relationships that may be relevant to cell-cell interactions, one may choose a neighborhood distance d up to 100 μm to reflect the limits of diffusion of epidermal growth factor that cells may use in paracrine signaling[30]. Additionally, visualizing the neighborhood may be used to guide the choice of d (Supp. Fig. 8a, c). For example, for the mouse cerebellum and embryo SRO datasets analyzed (Supp. Fig. 8a, b), we highlight how a neighborhood of 10 μm would be too small as it does not enclose a significant proportion of the cells given the density of cells in the tissues. On the other hand, a neighborhood of 100 μm would be too large as some of the cell types would incorporate all cells of other cell types inside the neighborhood buffer. Hence, a $d = 50$ μm was used for these SRO datasets. In general, the neighborhood distance should be chosen based on guidance from data visualization as well as biological prior knowledge.

Second, to create empirical null backgrounds of cell-type spatial relationships, CRAWDAD shuffles cell-type labels within non-overlapping tiles to create different null backgrounds. Although square tiles are used by default, hexagonal tiles are also available. To evaluate the robustness of trends given these different grid shapes, we created hexagonal tiles in our simulated dataset and repeated analysis (Supp. Fig. 9a). Comparing the Z-scores obtained at each scale on the different tiles, we noted a high correlation ($R = 0.99$) across all evaluated scales (Supp. Fig. 9b), suggesting the shape of the tiles is likely not a key factor in identifying spatial relationships, though both are available as options. The choice of the length scales should reflect the biological analysis of interest. However, choosing values smaller than the neighborhood distance may lead to trivial insignificant results.

Third, since CRAWDAD takes annotated cell types as input, the quality of the results directly depends on the quality of the annotation. Misannotated cell types could shift the proportions of other cell types inside spatial neighborhoods to alter the spatial relationships identified by CRAWDAD. Thus, cell-type annotations may be evaluated for robustness and cleaned if needed prior to CRAWDAD analysis[31,32]. Or alternatively, identified cell-type spatial relationships may be re-evaluated given multiple potential cell-type annotations to ensure the robustness of identified trends.

Finally, although we have elected to demonstrate CRAWDAD analysis on datasets from select SRO technologies, in general, CRAWDAD is amenable to any SRO technology for which spatial positions and associated labels can be derived. However, we caution that for some multi-cellular spot-based SRO technologies, additional deconvolution may be needed to ensure appropriate interpretation of results. In general, we recommend applying CRAWDAD to datasets with single-cell resolution to facilitate interpretation.

Overall, when used appropriately, such cell-type spatial relationship analysis enabled by CRAWDAD will provide another quantitative metric to facilitate the identification, characterization, and comparison of structural differences in tissues across axes of interest such as health and disease or development. Combined with the improvement in cell segmentation, we anticipate that future applications of spatial subsetting analysis such as that achieved with CRAWDAD can enable spatially-informed differential expression analysis to characterize subtle changes in cell state for cells of the same type colocalized within different microenvironment. Likewise, combined with other tools for identifying spatial niches or domains[33–35], we anticipate such cell-type spatial relationships may be characterized in a niche or domain-specific manner. Ultimately, we anticipate the analysis of SRO data with CRAWDAD can enable a more detailed quantitative characterization of cell-type spatial organization to contribute to our understanding of how spatial context and tissue architecture vary across conditions.

## Methods

### CRAWDAD overview

CRAWDAD characterizes cell-type spatial relationship trends by comparing observed pairwise cell-type spatial relationships to a set of empirical null distributions in which cell-type labels have been shuffled at different length scales.

**Creating null distributions at different length scales.** To generate empirical null distributions against which observed cell-type spatial relationships can be compared to evaluate for statistical significance, CRAWDAD employs a grid-based cell-type label shuffling strategy. Given a tissue containing cells represented by x-y spatial coordinates with cell-type annotations, we partition the tissue into non-overlapping side-by-side tiles. By default, tiles are squares of area $r^2$, where *r* is the size of the spatial length scale of analysis. Then for all cells that reside within the same square, cell-type labels are shuffled to create a null distribution for the given *r*. These shuffled null distributions are created for multiple *r* values to achieve a set of empirical null distribution at different length scales. We further create multiple permutations at each length scale by applying different random seeds and a grid-offsetting approach to mitigate the influence of spatial patterns that would benefit specific grid divisions. Specifically, the offsets are calculated by creating a sequence from 0 to *r*, in equally

spaced intervals of $r$ divided by the number of permutations. In each of the permutations, a different offset of the sequence will be applied. In addition to square tiles, CRAWDAD allows the creation of side-by-side non-overlapping hexagon tiles. In this case, the size of the scale is represented by the distance between opposite edges of the hexagon (Supp. Fig. 9a).

**Computing cell-type spatial relationship trends.** To evaluate the statistical significance of an observed pairwise cell-type spatial relationship, CRAWDAD uses a binomial testing framework. For a given reference cell type, CRAWDAD defines neighboring cells as those within a Euclidean distance of a user-defined neighbor distance d (default: 50 units) of any reference cell-type cells. CRAWDAD then calculates the proportion $p_1$ of neighboring cells that are members of a given query cell type:

$$p_1 = \frac{y_1}{n} \tag{1}$$

where n is the total number of total neighboring cells for a given reference cell type, and $y_1$ is the total number of cells of a given query cell type that are neighboring cells for a given reference cell type. By default, the reference cells, used to create the neighborhood, are removed from the proportion calculation.

This proportion $p_2$ of is also computed for the previously created null distributions:

$$p_2 = \frac{y_2}{n} \tag{2}$$

where $y_2$ is the total number of cells of the query cell type in the shuffled null distribution datasets that are inside the reference cell-type neighborhood calculated from the original data.

A two-sided binomial proportion test is then performed to test the equality of $p_1$ and $p_2$ against the alternative that they are not equal: $H_0 : p_1 = p_2$ versus $H_A : p_1 \neq p_2$. This is defined by a test statistic $Z$, which is:

$$Z = \frac{p_1 - p_2}{\sqrt{\frac{2p(1-p)}{n}}} \tag{3}$$

where:

$$p = \frac{y_1 + y_2}{2n} \tag{4}$$

**Defining the significance threshold.** We used a significance threshold of 5% ($p$-value < 0.05 or |Z-score|>1.96) corrected for multiple testing (Bonferroni method by default) based on the number of unique reference-neighbor cell-type pairs evaluated. In addition, when there were multiple permutations, we considered a Z-score significant if the mean value obtained across permutations was beyond the significance threshold.

**Summarizing and visualizing spatial relationship trends.** To visualize the spatial relationship results for a particular cell-type pair across different spatial extents, CRAWDAD uses trend plots where the x-axis represents the length scale and the $y$-axis represents the Z-score such that the trend represents how the spatial relationship of the cell-type pair changes as the spatial length scales increases. This visualization includes the Z-score threshold to determine significant relationships. When there are various permutations, the mean Z-score across all permutations is visualized alongside with an error bar of one standard deviation above and one bellow the mean.

To visually summarize results for all cell-type pairs, CRAWDAD creates a dot plot to show the length scale at which the spatial relationship trend for each cell-type pair first becomes statistically significant, encoded as the size of dots, as well as the associated Z-score at that scale, encoded as the color hue of dots. If a relationship is not significant, there will be no dot associated with it. Smaller scales are represented as larger dots to visually emphasize the potential importance of these small-scale colocalization relationships. Notably, such summaries do not fully capture the dynamics of pairwise cell-type spatial relationships as a function of length scale. Specifically, if a pairwise cell-type spatial relationship becomes significant at a scale of n, but not significant again at a scale of $m$ ($m > n$), only the n scale will be represented in the plot.

**Spatial subsetting of cells.** CRAWDAD can further assign cells of a given cell type to subsets based on the enrichment or depletion of cell types within their neighbors using its binomial testing framework. Given a tissue containing cells represented as x-y spatial coordinates with cell-type annotations, we again define a neighbor distance $d$. Given a reference and a query cell type, for every cell of the reference cell type we perform a one-sided binomial exact test to assess if the proportion of query cell type among the cell's neighbors are significantly greater than the proportion of the query cell type in the population. Cells of the given reference cell type with a p-value below a user-defined significance threshold can then be further subtyped as cells of the reference cell type that are enriched with cells of the query cell type.

## Creating, analyzing, and comparing simulated data across methods

**Simulating SRO data with manually defined spatial patterns.** To simulate a SRO dataset, we created 8000 cells randomly positioned from 0 to 2000 microns in the x and y coordinates. We chose four points $P = [(500, 500), (500, 1500), (1500, 500), (1500, 1500)]$ to be the centers of the circular neighborhoods defining the cell populations, where d(P) represents the distance from a cell to its closest point in P. We labeled the cells with d(P) ≤ 100 microns as cell-types B or C (50% chance each). Then, we labeled the cells with 100 <d(P) ≤ 300 microns as cell-type A. Finally, all the other cells were labeled cell-type D. For CRAWDAD, a set of shuffled null distribution was created at length scales of 100 to 1000 separated by 50 microns for 10 permutations. To identify significant trends with multiple testing correction, we used a Z-score threshold of 2.96.

To extend the simulated dataset and incorporate cell-type E, we copied the original cell positions, changed the labels to cell-type E, and placed one copy on the right, one on the top, and one on the top right of the original data. Therefore, the extended dataset has 32000 cells, 24000 from cell-type E, and ranges from 0 to 4000 microns in both x and y coordinates. We applied CRAWDAD to this dataset using the same parameters as those from the original simulated dataset analysis but increased the Z-score threshold to 3.09 to accommodate for multiple testing correction.

**Simulating SRO data using Gaussian random fields.** To create the simulated datasets with self-enrichment patterns, we followed the procedure previously described[11]. First, we simulate the position of 2000 cells by sampling from a uniform distribution ranging from 0 to 1 for both x- and y-axis. Then, we randomly split the cells in two groups of 1000 instances each. Using the Matern function with nugget variance of 0.1, shape parameter of 0.5, and smoothness parameter of 0.3, we created a covariance function to generate a Gaussian random field for each group. We binarize each field by assigning positive cells to one cell type and negative cells to the other. Finally, we merge both groups and scaled the cells' positions to 1000 microns. To generate all the ten datasets, we repeated this process using a different random seed for each.

**Comparing methods using simulated data.** To benchmark CRAW-DAD without relying on a significance threshold we performed a relative comparison between the relationship trends. For CRAWDAD, a set of shuffled null distribution was created at length scales of 100 to 500 separated by 50 microns for 10 permutations. We classified a cell type as enriched in the neighborhood of the reference cell type if it had the trend with the highest AUC value in the reference cell-type trend plot. Likewise, we classified a cell type as depleted if it had the most negative AUC value. We focused on measuring each method's capacity to distinguish trends, not the ability to identify statistically significant results.

### Analyzing real SRO data

**Analysis of the mouse cerebellum.** A pre-processed subset of a Slide-seqV2 dataset collected from an ~3000 μm-by-~2500 μm section of the mouse cerebellum was obtained from the original publication[4]. This dataset contained 10,098 beads with x-y coordinates and 19 cell-type annotations previously predicted by RCTD[14]. Poorly represented cell types defined as those being annotated in less than 20 beads (Choroid, Candelabrum, Ependymal, Globular, Macrophages) were not considered in the CRAWDAD cell-type colocalization, resulting in 14 remaining cell types. For interpretability, we converted the provided x-y coordinate units to micrometers by estimating the resolution values based on aligning the original publication's figure with annotated scale bars to the dataset coordinates. This led to an estimate of 0.64, which was multiplied to the x and y coordinate values of the original dataset to convert their units to micrometers.

For CRAWDAD, a set of shuffled null distributions were created at length scales of 100, 200, 300, 400, 500, 600, 700, 800, 900, 1000 microns for 10 permutations. A neighbor distance of 50 microns was used to evaluate every pairwise combination of cell types at each length scale. To identify significant trends with multiple testing correction, we used a Z-score threshold of 3.81.

**Analysis of the developing mouse embryo.** A pre-processed subset of the seqFISH data of an 8-12 somite stage embryo (Embryo 1) was obtained from the original publication[5]. This ~1000μm-by-1600μm dataset contained 19416 cells with x-y coordinates and 22 cell-type annotations. For interpretability, we converted the provided x-y coordinate units to micrometers by estimating the resolution values based on aligning the original publication's figure with annotated scale bars to the dataset coordinates. This led to an estimate of 1067.27784044, which was multiplied to the x coordinate values of the original dataset, and an estimate of 1578.21592795, which was multiplied to the y coordinate values of the original dataset to convert their units to micrometers.

For CRAWDAD, a set of shuffled null distributions was created at length scales of 100 to 1000 by 50 microns for 10 permutations. A neighbor distance of 50 microns was used to evaluate every pairwise combination of cell types at each length scale. To identify significant trends with multiple testing correction, we used a Z-score threshold of 3.88.

**Analysis of the human breast cancer.** We obtained a 7520.95μm-by-5471.17 μm Xenium breast cancer dataset (in situ sample 1, replicate 1) and with annotated cell types from the original publication[6]. We filtered the original data by removing cells with less than 3 gene counts, obtaining 162107 cells with x-y coordinates and 20 cell-type annotations.

For CRAWDAD, a set of shuffled null distributions was created at length scales of 100 to a 1000 by intervals of 100 microns for 3 permutations. A neighbor distance of 50 microns was used to evaluate every pairwise combination of cell types at each length scale. To identify significant trends with multiple testing correction, we used a Z-score threshold of 3.84.

**Analysis of the mouse brains.** We obtained the nine MERFISH mouse brain datasets from the Vizgen Data Release V1.0. May 2021[7] with cell types previously annotated through unified clustering[18]. We filtered the original data by removing cells with less than 3 gene counts and merging sub-cell types. The resulting number of cells and cell-type annotations by sample is provided in Supplementary Table 1.

For CRAWDAD, a set of shuffled null distributions was created at length scales of 100 to a 1000 by intervals of 100 microns for 3 permutations. A neighbor distance of 50 microns was used to evaluate every pairwise combination of cell types at each length scale.

**Analysis of the human spleen.** Pre-processed and compensated Akoya CODEX datasets of six human spleens tissue sections ranging from 3550.24μm-by-3423.43 μm to 4564.7μm-by-3423.43 μm in size from three different donors were downloaded from the HuBMAP Data Portal (https://portal.hubmapconsortium.org/) corresponding to dataset IDs: HBM389.PKHL.936, HBM772.XXCD.697, HBM556.KSFB.592, HBM825.PBVN.284, HBM568.NGPL.345, and HBM342.FSLD.938. These datasets contained protein expression for 28 markers, x-y coordinates, and cell segmentation area measurements for 154,446, 150,311, 152,896, 130,584, 177,777, and 226,384 segmented cells, respectively. Protein expression was normalized by cell area and $\log_{10}$ transformed with a pseudocount of 1. For interpretability, we converted the provided x-y coordinate units to micrometers using the resolution parameters provided by the HuBMAP Data Portal. We multiplied the x and y coordinate values of the original datasets by 0.3774038462 to convert their units from pixels to micrometers.

**Unified clustering and cell-type annotation.** Using dataset HBM389.PKHL.936, cells were clustered via Louvain graph-based clustering[36] based on a nearest neighbor-graph with k = 50. Protein expression values were summed together for cells assigned to each cluster, and then the values were scaled across protein expression measurements within each cluster. The scaled expression values were used to assign cell-type annotations, which initially resulted in 13 cell types and one unlabeled group. Outer and inner Follicular B cells annotations were combined into a single Follicular B cells annotation. Linear discriminant analysis (LDA) was performed to transfer cell-type labels between the datasets. Prior to label transfer, paired donor datasets HBM556.KSFB.592 and HBM568.NGPL.345, and HBM825.PBVN.284 and HBM342.FSLD.938 were harmonized[24] due to observed batch effects. LDA cell-type label transfers were done between the following reference and query datasets: HBM389.PKHL.936 −> HBM772.XXCD.697, HBM389.PKHL.936 −> HBM556.KSFB.592, HBM389.PKHL.936 −> HBM825.PBVN.284, HBM556.KSFB.592 −> HBM568.NGPL.345, HBM825.PBVN.284 −> HBM342.FSLD.938.

**CRAWDAD analysis of the human spleen.** For CRAWDAD, a set of shuffled null distributions were created at resolutions of length scales of 100 to 1750 by intervals of 50 microns for 10 permutations. A neighbor distance of 50 microns was used to evaluate every pairwise combination of cell types at each scale. The corrected Z-score threshold used was 3.58.

**Spatial subsetting of the human spleen.** CD4+ memory T cells were further subsetted based on their proximity to Follicle B cells using a one-sided exact binomial test, a neighbor distance defined as 50 microns and a p-value threshold of 0.05. With the subsets, we calculated the proportion of CD4+ memory T cells near Follicle B cells with respect to all CD4+ memory T cells and compared across datasets.

### Comparing across different samples

To compare different samples, we opted to use the AUC of the Z-score trend to represent each relationship instead of the scale of when the

relationship reaches significance as not all of them do. We represented each sample by the AUC values of each cell-type pair, creating an AUC high-dimensional space. By applying PCA to this space, we used the first two components to visualize the samples. In this case, instances that are similar in the high-dimensional space should also be similar in the low-dimensional one. Additionally, we investigated the variance of AUC across samples by plotting the variance for each cell-type pair in a dot plot. Lastly, we visualized the relationship trends for the cell-type pair with highest AUC variance across conditions.

### Analysis with other methods

**Ripley's Cross-K analysis.** Ripley's Cross K function draws a circular neighborhood around each reference cell, counts the number of cells of each type inside this region, and divide it by the cell-type density. This value is compared to the theoretical K. The multi-scale aspect of this analysis comes from varying the neighborhood size. Additionally, cells in the border of the tissue will consider areas that do not present any cell, requiring the application of border correction methods to mitigate this effect.

We used the spatstat (version 3.0-6) package[37] to compute different Ripley's Cross-K values for each pairwise combination of cell types. To compare with the theoretical K and perform border correction, we subtracted the theoretical K for a Poisson homogeneous processes from the isotropic edge corrected Ripley's Cross-K. For consistency in visualization, we set the maximum radius size to be the same as the maximum length scale evaluated in CRAWDAD.

**Squidpy's neighborhood enrichment.** We used Squidpy[10] (version 1.2.3) to apply its neighborhood enrichment implementation of the approach described by Schapiro et al.[9]. We defined the spatial neighbors using a radius of 50 and calculated Squidpy's neighborhood enrichment using default parameters. The results were plotted as a heatmap of Z-scores.

**Squidpy's co-occurrence probability.** Squidpy[10] implements the co-occurrence probability method originally presented in Tosti et al.[13]. The function works by drawing annular neighborhoods around each cell of the reference cell type. Then, it calculates the conditional probability of a cell type being enriched in that region. The multi-scale aspect of this analysis comes from varying the neighborhood size.

Using Squidpy (version 1.2.3) and its co_occurrence function with default parameters, we calculated the co-occurrence probability of clusters for each cell type. For consistency in visualization, we set the maximum distance to be the same as the maximum length scale evaluated in CRAWDAD.

### Reporting summary
Further information on research design is available in the Nature Portfolio Reporting Summary linked to this article.

## Data availability
All data analyzed with CRAWDAD is publicly available. The simulated datasets are available in CRAWDAD's Zenodo data repository[38] (https://doi.org/10.5281/zenodo.14004432). The Slide-seqV2 mouse cerebellum dataset was obtained from the original publication[4], with cell types previously annotated in RCTD[14], available at the Broad Institute Single Cell Portal at https://singlecell.broadinstitute.org/single_cell/study/SCP948. The seqFISH mouse embryo data was obtained from the original publication[5], available at https://doi.org/10.18129/B9.bioc.MouseGastrulationData. The Xenium human breast cancer dataset was obtained from the original publication[6], available at the GEO database under accession code GSE243280 https://www.ncbi.nlm.nih.gov/geo/query/acc.cgi?acc=GSE243280. The MERFISH mouse brain datasets were obtained from the Vizgen Data Release V1.0. May 2021[7], with cell types previously annotated in STalign[18], available at

https://doi.org/10.5281/zenodo.10724029. The CODEX human spleen samples were obtained from HuBMAP's data portal (https://doi.org/10.35079/HBM389.PKHL.936, https://doi.org/10.35079/HBM772.XXCD.697, https://doi.org/10.35079/HBM342.FSLD.938, https://doi.org/10.35079/HBM825.PBVN.284, https://doi.org/10.35079/HBM556.KSFB.592, https://doi.org/10.35079/HBM568.NGPL.345), with the cell type annotations performed in this paper available in CRAWDAD's Zenodo data repository[39] (https://doi.org/10.5281/zenodo.14004432). The source data files with the indication of which figures they relate to are provided in CRAWDAD's Zenodo data repository[38] (https://doi.org/10.5281/zenodo.14004432).

## Code availability
CRAWDAD is available as an open-source R package at https://github.com/jefworks-lab/crawdad, compressed and provided as Supplementary Software 1, with additional documentation and tutorials available at https://jef.works/CRAWDAD/[39]. Code to reproduce the analyses and results of this study is available on GitHub at https://github.com/rafaeldossantospeixoto/crawdad_revision_analysis[40].

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

## Acknowledgements

This material is based upon work supported by the HuBMAP Integration, Visualization, and Engagement (HIVE) Initiative under Award Number OT2-OD033760 (J.F., R.d.S.P., B.F.M., G.A.) and the National Institute of General Medical Sciences of the National Institutes of Health under Award Number R35-GM142889 (J.F., L.A., M.A.) and U54AI-142766 (M.A.B., M.A.A., T.M.B., C.H.W.). Parts of this work were carried out at the Advanced Research Computing at Hopkins (ARCH) core facility (rockfish.jhu.edu), which is supported by the National Science Foundation (NSF) grant number OAC1920103.

## Author contributions

B.F.M., R.d.S.P., and J.F. conceptualized the study. R.d.S.P. and B.F.M. developed the computational software. B.F.M. led data curation and initial explorations with support from L.A. and M.A. R.d.S.P. led the bioinformatics analyses. G.A. performed analyses and beta-tested software under the guidance of R.d.S.P. M.A.B. led the cell-type annotation of the spleen with assistance from B.F.M. M.A.B., M.A.A., T.M.B., and C.H.W. provided input to the methodology and contributed domain-specific biological expertize to the interpretation of the spleen results. R.d.S.P. and J.F. wrote the manuscript with initial drafts led by B.F.M. and J.F. All authors reviewed the results and approved the final version of the manuscript.

## Competing interests

The authors declare no competing interests.
