## [Transparent Peer Review file · Nature Communications]

Characterizing cell-type spatial relationships across length scales in spatially resolved omics data

Corresponding Author: Professor Jean Fan

Version 0:

Reviewer comments:

Reviewer #1

(Remarks to the Author)

In this manuscript, the authors proposed a method called CRAWDDAD that is designed to quantify cell-type spatial relationships across various length scales in spatial omics data. The study demonstrates the utility of CRAWDDAD in identifying and characterizing these relationships in both simulated and real data, providing insights into tissue organization and function. The manuscript is well-written and easy to follow. The work is of high impact potential to the field of computational biology and biomedical research in general.

I have only one major comment: a so-what question. CRAWDDAD provides a powerful approach to estimate the scale of co-localization. What's next: specifically, what can researchers do with this information in the next step? What types of subsequent analysis can this information be used?

Other comments:

1. The summary of relationship plot is not exactly symmetric, could the authors explain the reason?
2. The spatial gridding for creating the null distribution plays an essential role in the analysis. Could the authors explain more regarding the splitting: how exactly are the grids split and do the squares overlap with each other? The grids are squares, which is a rather natural choice. But does the shape of the grid also affect the power? Along the line, does the shape of spatial pattern affect the test power?
3. How did authors evaluate type-I error?
4. It seems that the authors fix d (spatial distance for selecting neighbors) and change r to determine the scale. How to determine d in real data analysis?
5. The test score may be affected by cell numbers in a local region. How did authors adjust for this kind of spatial imbalance in cell density?
6. Similarly, when cell type A has two types of neighbors: cell type B and cell type C, but the number of cells of type B is much larger than that of type C. The spatial colocalization pattern of cell type C will be diluted, how did authors adjust for this?
7. Does CRAWDDAD work for spot level data?
8. It would nice to see a comparison with the following approach: first using spatial grid to group cells and then calculating the correlation between cell proportion of two cell types in different distance scale (keep only grids that contain two cell types).
9. For Figure 1c, why authors use large circle for 350 μm and smaller circles for 550 μm ?

Reviewer #2

(Remarks to the Author)

This manuscript authored by Jean's group describes CRAWDDAD, a tool designed to quantify spatial relationships between cell types across different length scales. The authors illustrate the utility of their tool using three datasets, encompassing the brain, spleen, and embryo. My comments are as follows.

Major:

When it is first introduced in the manuscript, please provide a clear definition of "length scale" from both biological and

computational perspectives.

As 10x Visium is the predominant platform in the market, it is essential to demonstrate the method's compatibility with this data. Moreover, does the method align with the availability of various deconvolution (<https://doi.org/10.1038/s41467-023-40458-9>, <https://doi.org/10.1038/s41587-022-01273-7>) and super-resolution (<https://doi.org/10.1038/s41587-021-00935-2>, [10.1016/j.cels.2023.03.008](https://doi.org/10.1016/j.cels.2023.03.008)) methods?

The dataset for demonstration is limited to only three platforms, which is relatively small compared to the wide array of spatial platforms currently available. To achieve a wider scope for an interdisciplinary journal like Nat Comm, additional spatially resolved techniques, such as MIBI-TOF, nanostring-SMI, and MERSCOPE, etc., should be included in the analysis.

The manuscript primarily focuses on highly organised tissues. It is vital to demonstrate how the method can be applied to tissues with less organisation, such as tumours and other complex diseases, to assess its general applicability.

Comparing colocalisation summaries between different treatment/condition/perturbation scenarios holds greater significance than evaluating a single tissue in isolation. The manuscript should provide further elaboration on how the method can be adjusted to derive statistical power in these comparative scenarios.

The definition of "cell type" significantly influences the method's output (e.g., inaccurate cell type identifications and granularity determination, etc.). The method can benefit from considering: (1) the cell type continuum in the gene expression space; (2) cases where cell typing is not perfectly accurate. This issue becomes particularly pronounced in cases such as Figure 3b, where the cell type boundary in the feature space is unclear.

The simulation is overly simplified and can be improved by designing multiple simulation studies, considering (1) a continuous cell state along the gene expression space, (2) mixed cell types in space, (3) inaccurate cell type identification, (4) noise in gene expression (Gaussian, Poisson), and (5) uneven spatial distributions.

We found the case study on Slide-seq data in Figure 2 unreliable. Slide-seq (V1/V2) is known to capture parts of multiple or partial cells for each spot, leading to unrealistic cell type and spatial localisation. It appears that the proposed method does not consider this.

Furthermore, for the Slide-seq case, it is valuable to see the proposed method's output align with known biology. However, for a method paper, there should be a quantitative comparison between the proposed method and others.

In Figure S2d-f, it is unclear why CRAWDAD is superior to Squidpy and Ripley based on the figure. The author should clearly define what the golden standard ground truth is for each dataset to determine which method is better. Similar issues arise in Figure S2g-l and Figure S2j-l. Please establish a more solid and comprehensive benchmarking analysis in the manuscript.

A comparison of running time and memory usage against other related tools is necessary.

I am not sure if this be necessary element for a Nat Comm paper. Unique biological insights brought by the proposed method are lacking. The author should demonstrate how, by using CRAWDAD, individuals can identify previously unknown biological discoveries that cannot be identified by similar tools (such as Squidpy).

Minor:

Consider splitting complex sentences, such as Line 44-48, into multiple sentences for improved clarity.

Without substantial improvement, I believe this work seems to be immature for Nature Communications in terms of technical novelty, simulation design, rigor of benchmarking, and biological insights, particularly when compared with other Nature Communications papers on similar topics.

Version 1:

Reviewer comments:

Reviewer #1

(Remarks to the Author)

The authors have carefully addressed my critiques. I have no additional comments.

(Remarks on code availability)

Reviewer #2

(Remarks to the Author)

I commend the significant efforts made to address all my previous comments. Now I support the publication of this work and look forward to the new biological insights that this tool will bring.

(Remarks on code availability)

We sincerely thank the editor and the reviewers for their insightful and constructive feedback in helping us improve this manuscript. We have now revised the manuscript to address all the points raised by the reviewers, organized herein as a point-by-point response. Throughout this point-by-point response, reviewer comments are shown in **blue**, with our responses in **green**, and changes to the manuscript in **black**.

Reviewer #1

We thank the reviewer for their helpful feedback. To thoroughly address these points, we have opted to rewrite the manuscript as a full Article instead of a Brief Communication. We hope the longer length format will clarify these points for readers.

Major Comments

1. I have only one major comment: a so-what question. CRAWDAD provides a powerful approach to estimate the scale of co-localization. What's next: specifically, what can researchers do with this information in the next step? What types of subsequent analysis can this information be used?

We thank the reviewer for the great question. Beyond estimating the scale of cell-type co-localization, we have now added additional analyses to demonstrate how CRAWDAD can be used to compare such cell-type co-localization trends across tissues in different conditions. Such information can give insight into inter- and intra-individual variation linked to donor and tissue-specific features.

Briefly, we used CRAWDAD to compare nine MERFISH mouse brain samples and show that samples from the same brain location have more similar cell-type spatial relationships than those from different regions. We have added these results to the revised manuscript, provided below for the reviewer's convenience:

Lines 246-275:

Beyond characterizing cell-type spatial relationships within a single sample, such multi-scale characterization enabled by CRAWDAD can also be used to compare cell-type spatial relationships across samples spanning different conditions, such as health and disease, development, or replicates. To demonstrate this functionality, we applied CRAWDAD to nine mouse brain samples assayed by MERFISH comprised of three replicates from three distinct Bregma locations⁷ with cell-type annotations obtained previously using unified clustering¹⁸ (Fig 4a). We applied CRAWDAD to evaluate 734,693 annotated cells across all datasets representing 14 cell-types using a neighborhood size of 50 μ m across length scales ranging from 100 μ m to 1000 μ m (Methods). To compare multi-scale cell-type spatial relationships across samples, we calculated the signed area under the curve (AUC) for each Z-score trend for each cell-type pair. We then performed dimensionality reduction with principal component analysis (PCA) on all signed AUC values to find that cell-type spatial relationships of replicates from the same Bregma location are highly similar, as they are positioned closer together in PC space (Fig 4b). We likewise overall observed a smaller variance in the signed AUC values within replicates from the same

Bregma location compared to across locations (Fig 4c). These results suggest that samples from the same Bregma location have cell-type spatial relationships that are more similar than those from different Bregma locations, as expected. Importantly, this similarity in cell-type spatial relationships is robust to tissue rotation and small local diffeomorphisms, as some of the brain tissue sections profiled are rotated with small tissue distortions compared to others. To investigate specific highly variable cell-type spatial relationships further, we visualized the spatial-relationship trends for the cell-type pair with the highest signed AUC variance across locations: GABAergic Estrogen-Receptive Neurons as reference and Excitatory Neurons as neighbor (Fig 4d). Despite its comparatively higher signed AUC variance across locations, samples from the same Bregma location still generally exhibited the same depletion trend whereas samples across Bregma locations varied (Fig 4d). Visual inspection of GABAergic Estrogen-Receptive Neurons and Excitatory Neurons also suggested high consistency in terms of spatial relationships within replicates from the same Bregma location compared to across locations (Fig 4e). As such, cell-type spatial relationship trends quantified by CRAWDAD can be used to compare across samples to confirm that cell-type spatial relationships in the mouse brain are generally highly consistent within replicates from the same Bregma location compared to across locations.

Figure 4. CRAWDAD enables comparison of spatial relationships across different tissue sections of the mouse brain assayed by MERFISH. a. Spatial visualization of annotated cell types in each sample. Scale bars correspond to 1000 μ m. b. Visualization of each sample in the principal component space with the first two principal components calculated using the standardized AUC

of each multi-scale cell-type spatial relationship trend. c. Variability of multi-scale cell-type spatial relationship trends calculated as the variance of the AUC values across replicates (top) and locations (bottom). d. The multi-scale cell-type spatial relationship trend plot of GABAergic Estrogen-Receptive (ER) Neurons as reference and Excitatory Neurons as neighbor for samples of replicates from the same location (top) and different locations (bottom). e. Spatial visualization of the GABAergic ER Neurons and Excitatory Neurons in replicates from the same location (left) and different locations (right).

We have also described these datasets in the methods section:

Lines 589-597:

Analysis of the mouse brains

We obtained the nine MERFISH mouse brain datasets from the Vizgen Data Release V1.0. May 2021⁷ with cell types previously annotated through unified clustering¹⁸. We filtered the original data by removing cells with less than 3 gene counts and merging sub-cell types. The resulting number of cells and cell-type annotations by sample is provided below:

Location	Replicate	Number of Cells	Number of Cell Types	Tissue Size
1	1	78329	13	9036.87x 6326.88
	2	88884	13	8060.09x 9936.43
	3	84635	14	8504.56x 8249.1
2	1	83546	14	8883.69x 7113.93
	2	84171	14	8867.34x 9316.67
	3	85957	14	9147.76x 6980.63
3	1	70844	14	7058.34x 7829.3
	2	83461	14	8952.54x 6747.24
	3	74866	14	8952.54x 6747.24

For CRAWDAD, a set of shuffled null distributions were created at length scales of 100 to a 1000 by intervals of 100 microns for 3 permutations. A neighbor distance of 50 microns was used to evaluate every pairwise combination of cell-types at each length scale.

Lines 636-645:

Comparing across different samples

To compare different samples, we opted to use the area under the curve (AUC) of the Z-score trend to represent each relationship instead of the scale of when the relationship reaches significance as

not all of them do. We represented each sample by the AUC values of each cell-type pair, creating an AUC high-dimensional space. By applying principal component analysis to this space, we used the first two components to visualize the samples. In this case, instances that are similar in the high-dimensional space should also be similar in the low-dimensional one. Additionally, we investigated the variance of AUC across samples by plotting the variance for each cell-type pair in a dot plot. Lastly, we visualized the relationship trends for the cell-type pair with highest AUC variance across conditions.

Additionally, we used the same approach to compare the different human spleen samples. We observed that most spatial cell-type relationships are highly consistent (low variability) across all samples. However, some spatial cell-type relationships were patient-specific, while others will vary across patients and samples. We have added these results to the revised manuscript, provided below for the reviewer's convenience:

Lines 298-313:

To determine whether such cell-type spatial colocalization relationships are consistent across tissue sections and individuals, we further repeated these analyses with 837,952 cells from five additional spleen samples both within and across individuals. To ensure all datasets were annotated in a uniform manner, we applied batch-correction²⁴ and used a linear discriminant analysis model to transfer cell-type annotations to these new datasets (Methods, Supp Fig 5). We then applied CRAWDAD to identify similar cell-type spatial relationships corresponding to the WP and RP compartments both within and across individuals (Fig 5d, Supp Fig 6a). Further analyzing the variance of the relationship trend's AUC values (Fig 5e), we noticed that most cell-type spatial relationship trends were highly consistent across patients and samples, reflecting the ordered patterning of the functional tissue regions (Fig 5e-f, Supp Fig 7a). Select cell-type spatial relationship trends had patient-specific relationships, exhibiting consistent trends within replicates from the same patient but varying across patients, suggestive of potential patient-specific variation (Fig 5g, Supp Fig 7b). Other cell-type spatial relationship trends varied even within replicates, suggestive of potential tissue sample-specific patterns (Fig 5h, Supp Fig 7c). In general, we anticipate assessing these variations in cell spatial relationships can give insight into inter- and intra-individual variation linked to donor and tissue-specific features.

Figure 5. CRAWDAD characterize cell-type spatial relationships in the human spleen assayed by CODEX. a. Heatmap of marker protein expression for annotated cell types. b. UMAP reduced-dimensional visualization of annotated cell-types. c. Spatial visualization of annotated cell-types in one representative tissue section. Scale bars correspond to 250 μ m. d. Summary visualization of the multi-scale cell-type spatial relationship analysis for tissue sections PKHL and XXCD from patient HBM966.VNKN.965. Cell types consistently colocalized in the white and red bulk are highlighted with small and large squares respectively. e. Variability of multi-scale cell-type spatial relationship trends calculated as the variance of the AUC values across samples. The multi-scale cell-type spatial relationship trend plots are shown for select cell-type pairs exhibiting f. low variability across different samples, g. high variability across patients but low variability within replicates, and h. high variability across samples including within patients. i. Subset of CD4+ Memory T cells near Follicle B cells. The number of CD4+ Memory T cells (n) and the proportion of subsets (left) and spatial visualization of subsets (right) in tissue sections PKHL and XXCD from patient HBM966.VNKN.965. j. Proportion of CD4+ Memory T cells near Follicle B cells over all CD4+ Memory T cells in each sample.

Finally, we have also extended the discussion to include this and further elaborate on what researchers may be able to do with this information in the next step both in terms of subsequent analyses as well as potential experiments. In particular, we emphasize that CRAWDAD's spatial subsetting enables additional spatial differential analysis between cells from the same type based on their enrichment or depletion around other cell types. We have provided relevant excerpts to the discussion below for the reviewer's convenience:

Lines 400-411:

Overall, when used appropriately, such cell-type spatial relationship analysis enabled by tools like CRAWDAD will provide another quantitative metric to facilitate the identification, characterization, and comparison of structural differences in tissues across axes of interest such as health and disease or development. Combined with the improvement in cell segmentation, we anticipate that future applications of spatial subsetting analysis such as that achieved with CRAWDAD can enable spatially-informed differential analysis to characterize subtle changes in cell state for cells of the same type colocalized within different microenvironment. Likewise, combined with other tools for identifying spatial niches or domains³³⁻³⁵, we anticipate such cell-type spatial relationships may be characterized in a niche or domain-specific manner. Ultimately, we anticipate the analysis of SRO data with CRAWDAD can enable a more detailed quantitative characterization of cell-type spatial organization to contribute to our understanding of how spatial context and tissue architecture vary across health, disease, and development.

Minor Comments

1. The summary of relationship plot is not exactly symmetric, could the authors explain the reason?

The reviewer is correct that the relationship dot plot is not symmetric. Briefly, there are two possible reasons for this:

- One possibility is due to an imbalance in the cell type locations. In this case, cells from one type are close to some cells of another type but not to all of them. As an example, in the

cerebellum, Supp Fig 4 a-d shows this phenomenon in the relationship of UBCs and granule cells – UBCs are always close to granule cells but there are also some granule cells not close to UBCs.

- The second possibility is caused by an imbalance in the cell type density. One cell type can have cells largely concentrated in one region, with a few dispersed elsewhere. In this case, the dispersed cells will contribute to the creation of the neighborhood that will consider the cells around them. However, due to their small number, they will not significantly contribute to the proportions as a neighbor cell type. In the embryo, Supp Fig 4 e-h shows this relationship between presomitic mesoderm and spinal cord cells – presomitic mesoderm cells are largely concentrated outside of the spinal cord cells' neighborhood, but a few presomitic mesoderm cells are present inside it.

We have added an explanation of this asymmetry to the results section with a supplementary figure, provided below for the reviewer's convenience:

Lines: 223-243

In general, we note that the cell-type spatial relationships identified in CRAWDAD are not always symmetric. Asymmetric results may be caused by two scenarios: location imbalance and density imbalance. In location imbalance, cells of the neighboring cell-type may be close to only some cells of the reference cell-type, but not all. For example, the neighborhood of UBCs is enriched with granule cells (Supp Fig 4a, c). However, UBCs are rare and present in only a small proportion of the granule cells' neighborhood and therefore does not represent a significant relationship (Supp Fig 4b, d). In density imbalance, cells from one type are highly concentrated in one region, with a few dispersed across other parts of the tissue. Therefore, the sparse cells will contribute to the creation of the neighborhood as the reference cell type but will not significantly contribute to the proportions as the neighbor cell type, due to their small number. For example, a large part of the presomitic mesoderm's neighborhood is created by its sparse cells, which encapsulate spinal cord cells, creating a relationship of enrichment (Supp Fig 4e, g). On the other hand, most of the presomitic mesoderm cells are outside the spinal cord's neighborhood creating a relationship of depletion (Supp Fig 4f, h). Such asymmetric cell-type spatial relationships may reflect non-exclusive cell-type interactions. For example, immune cells may infiltrate a focal tumor such that the neighborhood of tumor cells will be enriched with immune cells, but the neighborhood of immune cells might not be enriched by tumor cells given their widespread spatial distribution throughout the body, consistent with a non-exclusive cell-type spatial relationship at a whole-body spatial extent¹⁷. Therefore, CRAWDAD can quantitatively capture such asymmetric cell-type spatial relationships and effectively delineate cell-type spatial relationships across multiple length scales for diverse tissues and SRO technologies.

Supplementary Figure 4. Sample asymmetric cell-type spatial relationships. Spatial visualization of cells in the cerebellum with the neighborhood of UBCs (a) and Granule cells (b) outlined. CRAWDAD's multi-scale spatial relationship trend plot for UBCs and Granule cells with UBCs as the reference cell-type and Granule cells as the neighboring cell-type (c) and vice versa (d). Spatial visualization of cells in the embryo with the neighborhood of presomitic mesoderm cells

(e) and spinal cord cells (f) outlined. CRAWDAD's multi-scale spatial relationship trend plot for presomitic mesoderm cells and spinal cord cells with presomitic mesoderm cells as the reference cell-type and spinal cord cells as the neighboring cell-type (g) and vice versa (h). Scale bars correspond to 250 μ m.

2. The spatial gridding for creating the null distribution plays an essential role in the analysis. Could the authors explain more regarding the splitting: how exactly are the grids split and do the squares overlap with each other? The grids are squares, which is a rather natural choice. But does the shape of the grid also affect the power? Along the line, does the shape of spatial pattern affect the test power?

We thank the reviewer for the opportunity to clarify this point. In summary, we create the grids by dividing the tissue into side-by-side non-overlapping squares with sides of the scale size. In addition, we allow the apply to use different permutations to accommodate for spatial patterns that might benefit from a perfect alignment with the grids.

We have now added a clarification of this to the methods, provided below for the reviewer's convenience:

Lines 423-438:

Creating null distributions at different length scales

To generate empirical null distributions against which observed cell-type spatial relationships can be compared to evaluate for statistical significance, CRAWDAD employs a grid-based cell-type label shuffling strategy. Given a tissue containing cells represented by x-y spatial coordinates with cell-type annotations, we partition the tissue into non-overlapping side-by-side tiles. By default, tiles are squares of area r^2 , where r is the size of the spatial length scale of analysis. Then for all cells that reside within the same square, cell-type labels are shuffled to create a null distribution for the given r . These shuffled null distributions are created for multiple r s to achieve a set of empirical null distribution at different length scales. We further create multiple permutations at each length scale by applying different random seeds and a grid-offsetting approach to mitigate the influence of spatial patterns that would benefit specific grid divisions. Specifically, the offsets are calculated by creating a sequence from 0 to r , in equally spaced intervals of r divided by the number of permutations. In each of the permutations, a different offset of the sequence will be applied. In addition to square tiles, CRAWDAD allows the creation of side-by-side non-overlapping hexagon tiles. In this case, the size of the scale is represented by length of the sides of the hexagon (Supp Fig 9a).

We agree with the reviewer that square grids are a natural choice. However, we acknowledge that, particularly given more circular tissue structures, hexagonal grids may also be a reasonable option. We have therefore updated CRAWDAD to allow users to choose hexagonal grids as an alternative. To determine the potential impact of the shape of the grid, we evaluated the spatial enrichments/depletions Z-scores across resolutions and cell-type pairs in our simulated data with either square or hexagonal grids. We find highly correlated z-scores across resolutions, suggesting that the shape of the grid does not substantially impact results. We have now included these results in discussion with a supplementary figure, provided below for the reviewer's convenience:

Lines 377-384:

Second, to create empirical null backgrounds of cell-type spatial relationships, CRAWDAD shuffles cell-type labels within non-overlapping tiles to create different null backgrounds. Although square tiles are used by default, hexagonal tiles are also available. To evaluate the robustness of trends given these different grid shapes, we created hexagonal tiles in our simulated dataset and repeated analysis (Supp Fig 9a). Comparing the Z-scores obtained at each scale on the different tiles, we noted a high correlation ($R = 0.99$) across all evaluated scales (Supp Fig 9b), suggesting the shape of the tiles is likely not a key factor in identifying spatial relationships, though both are available as options.

Supplementary Figure 9. a. Visualization of the shuffled labels using the square and hexagonal grid tiles on simulated dataset for finer and coarser scales. b. Consistency of Z-scores and scales for the different grid tile shapes. The x-axis represents the Z-score obtained using the square grid tiles and the y-axis represents the Z-score obtained using the hexagon grid tiles. The color saturation represents the scale in which the value was obtained. The red line represents the correlation of the values.

3. How did authors evaluate type-I error?

We thank the reviewer for bringing up this important question. To perform a quantitative evaluation of the methods, we created ten simulated datasets with self-enrichment patterns using a simulation approach previously described [Viladomat et al, *Biometrika* 2014]. Briefly, we used a Matern autocorrelation function to create two independent gaussian random fields. We binarize the gaussian random field to create two spatially separated cell-types. Due to the autocorrelation, each cell-type should be colocalized with itself and separated from the other cell type originated in the same gaussian field. Likewise, given the underlying independent gaussian random fields, cell-type pairs across the two independent instantiations should exhibit no spatial relationship. We repeated this process to generate ten datasets.

Then, we evaluated the ability of CRAWDAD, Ripley's K and Squidpy in distinguishing enrichment and depletion between cell types. As Ripley's K and Squidpy's co-occurrence methods do not present a statistical threshold to detect significant relationships, to perform a fair comparison between our method and the others, we determined the cell-type relationships based on a relative comparison of the Z-score trends. The neighbor cell type which Z-score trend had the highest area under the curve (AUC) value the was considered enriched around the reference cell type, while the one with the lowest AUC value was considered depleted. Since there are only two true relationships for each cell type in our simulated datasets, this strategy is able to detect them even though it does not rely on a statistical threshold. As such, we considered detecting the relationship (enrichment or depletion) as a positive result and not detecting it as a negative result and calculated the number of false positives for each tool. Based on this simulation framework, we obtained a true positive rate of 0.95 for CRAWDAD, 0.86 for Squidpy's co-occurrence implementation, and 0.8 for Ripley's K Cross.

We have now added the results for these simulations in the revised manuscript with relevant excerpts provided below for the reviewer's convenience:

Lines 140-175:

To further benchmark and compare CRAWDAD's functionality, we simulated a variety of SRO datasets using a previously developed simulation framework¹¹ (Methods). Briefly, we simulated cells by sampling from a uniform distribution to create x-y positions. We split the cells into two groups, and, for each group, we associated a value to each cell using independent, autocorrelated Gaussian random fields (Supp Fig 2a). We binarized the values, splitting the cells into two cell-types based on the underlying simulated value (Supp Fig 2b). In this manner, we created a simulated dataset with four cell types (Supp Fig 2c) where we expect each cell-type to be enriched with itself due our use of spatially autocorrelated simulation values. Likewise, we can expect the two cell-types simulated from the same Gaussian random field to be spatially mutually exclusive and therefore identified to be separated. Additionally, we expect the cell-types from different random fields to exhibit no significant spatial relationship due to the Gaussian random fields being independent. We repeated this process to create a total of ten random simulated datasets.

We used these simulated datasets to benchmark and compare CRAWDAD with two other spatial relationship analysis methods that also consider spatial length scales, Ripley's K Cross¹² and Squidpy's co-occurrence implementation of the approach described in Tosti et al.¹³ Although all evaluated methods perform cell-type enrichment analysis across length scales, their definition of length scales differs. Briefly, Ripley's K Cross evaluates multiple length scales by increasing the neighborhood size while comparing the cell-type proportion in the neighborhood to the global proportion. On the other hand, Squidpy's co-occurrence implementation evaluates multiple length scales by increasing the size of an annulus neighborhood and calculating the conditional probability of the neighbor cell types given the reference cell type. In addition, as Ripley's K Cross and Squidpy's co-occurrence implementation do not present a threshold to determine statistical significance, for comparative purposes, we opted to assess each method's ability to distinguish between cell-type spatial enrichment and depletion. Specifically, given a reference cell type, we considered a method as achieving a true positive prediction if the cell type identified with the most enriched relationship trend was itself. Alternatively, we also considered a method as achieving a true positive prediction if the cell type identified with the most depleted relationship trend was the

other cell type from the same Gaussian random field. We identified the cell-type with the most enriched and most depleted spatial trend using their area under the trend curve value (Methods). Using this approach, we evaluated all four cell types across all ten simulated datasets using all three methods (Supp Fig 2d). Based on this simulation framework, we obtained a true positive rate of 0.95 for CRAWDAD, 0.86 for Squidpy's co-occurrence implementation, and 0.8 for Ripley's K Cross. In this manner, cell-type spatial relationships identified by CRAWDAD can more accurately distinguish between cell-type spatial enrichment and depletion compared to other evaluated methods based on simulated data.

Supplementary Figure 2. Creation of the simulated dataset. a. Two independent Gaussian random fields with uniformly sampled spatial positions representing 1000 cells. b. These spatial positions are assigned a cell type based on the random field's value. c. Both fields are combined to generate one dataset with 4 cell-types. d. Relationships trends obtained by each method using cell-type A as the reference cell-type.

Additionally, we've elaborated on how the simulation was created in the methods, provided below for the reviewer's convenience:

Lines 536-545:

Simulating SRO data using Gaussian random fields

To create the simulated datasets with self-enrichment patterns, we followed the procedure previously described¹¹. First, we simulate the position of 2000 cells by sampling from a uniform distribution ranging from 0 to 1 for both x and y axis. Then, we randomly split the cells in two groups of 1000 instances each. Using the Matern function with nugget variance of 0.1, shape parameter of 0.5, and smoothness parameter of 0.3, we created a covariance function to generate a Gaussian random field for each group. We binarize each field by assigning positive cells to one cell type and negative cells to the other. Finally, we merge both groups and scaled the cells' positions to 1000 microns. To generate all the ten datasets, we repeated this process using a different random seed for each.

We also explain how the methods were compared in the methods, provided below for the reviewer's convenience:

Lines 547-553:

Comparing methods using simulated data

To benchmark CRAWDAD without relying on a significance threshold we performed a relative comparison between the relationship trends. We classified a cell-type as enriched in the neighborhood of the reference cell type if it had the trend with the highest area under the curve (AUC) value in the reference cell-type trend plot. Likewise, we classified a cell type as depleted if it had the most negative AUC value. We focused on measuring each method's capacity to distinguish trends, not the ability to identify statistically significant results.

4. It seems that the authors fix d (spatial distance for selecting neighbors) and change r to determine the scale. How to determine d in real data analysis?

We thank the reviewer for bringing up this great point. In a biological setting, we generally choose d based on known biological constraints. For example, the limit of diffusion of small molecules that cells may use to communicate and interact with each other is generally between 10 to 100 μm . As such, we may choose $10 < d < 100\mu\text{m}$ based on such known effective paracrine signaling distances.

In practice, visualization is quite helpful in choosing an appropriate d . Taken to the extreme, if d is too small (smaller than the width of 1 cell), there would not be other cells inside the neighborhood, resulting in NaN Z-scores. Alternatively, if d is too large, every cell would be inside the neighborhood and the cell-type proportions would stay the same after shuffling, leading no significant relationships.

We have updated our schematic in Figure 1 to better clarify this important distinction between d and length-scale, provided below for the reviewer's reference.

Figure 1. Motivating Cell-type Relationship Analysis Workflow Done Across Distances (CRAWDAD) using simulated data. a. Illustration of the cell-type spatial relationships found at different length scales. b. Simulated spatial omics tissue data, visualized at different scales. Each point is a cell, colored by cell type. Scale bars correspond to 250 μm . c. Representation of the creation of the neighborhood and the null background. CRAWDAD draws a circle (neighborhood distance as the radius) around each cell of the reference cell type and merges them into one

neighborhood. d. CRAWDAD creates a grid of side-by-side tiles (length scale defined as the side length for square pixels and the distance between opposite edges for hexagonal pixels) and shuffles the labels inside each tile to create the null background. e. The multi-scale spatial relationship trend plot for reference cell-type C and neighbor cell-type B. The horizontal dotted lines represent the z-score significance threshold corrected for multiple testing ($Z\text{-score} = \pm 2.96$). The red bars denote the standard error for the z-score estimated using permutations. f. The multi-scale spatial relationship trend plot for reference cell-type A and neighbor cell-type B. g. Summary visualization all cell-type spatial relationships. The size of the dot represents the scale in which a neighbor cell-type first reaches a significant spatial relationship with respect to a reference cell-type. The color of the dot is the z-score at such scale. Created in BioRender. Fan, J. (2023) BioRender.com/y47n964.

In addition, we have now incorporated a function in CRAWDAD to aid the choice of d , added more information about the choice of d , and included a new supplementary figure to visualize its application:

Lines 355-376:

Although we have demonstrated CRAWDAD to be a potentially useful tool in identifying, characterizing, and comparing cell-type spatial relationships, there are several considerations worth noting as they may influence interpretation. First, CRAWDAD results rely on a few user-defined parameters. In particular, it uses a fixed neighborhood distance d to determine the size of the neighborhood used to consider neighboring cells. In the context of geospatial analysis, such sensitivity of results to the neighborhood distance has been previously characterized as the sensitivity to kernel bandwidth²⁹. We note that if the defined d is too small, the neighborhood will only contain cells from the reference cell type. In such a scenario, the total number of neighbor cells would be zero, leading to non-significant results. Alternatively, if d is too large, the neighborhood will encompass all the cells in the sample. In this case, the proportions of cell types within the neighborhood before and after shuffling will remain the same, leading to non-significant results. Generally, we recommend choosing d based on the biological constraints of the analysis. For example, to identify cell-type spatial relationships that may be relevant to cell-cell interactions, one may choose a neighborhood distance d up to 100 μm to reflect the limits of diffusion of epidermal growth factor that cells may use in paracrine signaling³⁰. Additionally, visualizing the neighborhood may be used to guide the choice of d (Supp Fig 8). For example, for mouse cerebellum and embryo SRO datasets analyzed, we highlight how a neighborhood of 10 μm would be too small as it does not enclose a significant proportion of the cells given the density of cells in the tissues. On the other hand, a neighborhood of 100 μm would be too large as some of the cell types would incorporate all cells of other cell types inside the neighborhood buffer. Hence, a $d = 50 \mu\text{m}$ was used for these SRO datasets. In general, the neighborhood distance should be chosen based on guidance from data visualization as well as biological prior knowledge.

Supplementary Figure 8. The effects of the neighborhood size. a-c. Histogram of the cell-type proportion of cells from the neighbor of each cell type given neighborhood sizes of 10, 50 and 100 μm (top). Corresponding spatial visualization of the neighborhood as a black outline for chosen cell type (bottom). (a) Visualization of the cerebellum proportions and spatial visualization Granule neighborhoods. (b) Visualization of the embryo proportions and spatial visualization of the Endothelium neighborhoods. (c) Visualization of the PKHL spleen proportions and spatial visualization of the Neutrophils/Monocytes neighborhoods. Scale bars correspond to 250 μm .

5. The test score may be affected by cell numbers in a local region. How did authors adjust for this kind of spatial imbalance in cell density?

We appreciate the reviewer's comment and agree that spatial density will affect the results. Notably, in our statistical evaluation, we consider the number of cells in each reference cell-type's neighborhood. For some cell-types, this neighborhood may have lower cell density. However, because we use a permutation-based approach, this lower cell density is maintained in the permuted null.

6. Similarly, when cell type A has two types of neighbors: cell type B and cell type C, but the number of cells of type B is much larger than that of type C. The spatial colocalization pattern of cell type C will be diluted, how did authors adjust for this?

We thank the reviewer's comment and acknowledge that rare cell types can affect CRAWDAD results. To accurately quantify the statistical significance of cell-type spatial relationships, CRAWDAD relies on an accurate quantification of cell numbers for all cell-types represented in a tissue. As presented in the methods, the Z-score formula not only considers the proportions before and after shuffling, but also total number of cells inside the neighborhood. Consequently, rare cell types will generally have smaller Z-scores both as neighbor and as reference. As the reference cell-type, rare cell types may have smaller neighborhoods, which will enclose a small number of cells inside, contributing to a lower Z-score. As the neighboring cell-type, rare cell types will have fewer cells inside the neighborhood, requiring a greater difference in the proportions to achieve statistical significance. These results are expected as it demands greater sample sizes to achieve higher significance values. Normalizing by the number of cells would create bias towards rare cell types and imply more confidence in observations that are not truly present. We anticipate SRO technologies that selectively profile a subset of cells in a tissue such as Slide-tags [Russel et al, Nature 2024] will require additional characterization to ensure accurate results with CRAWDAD analysis.

7. Does CRAWDAD work for spot level data?

We appreciate the reviewer's question. As CRAWDAD only considers spatial positions and their associated labels, technically, it could be applied to spot-based SRO. However, since spot-based SRO data may have spots with multiple cells, the results should be interpreted with care. We theorize that to properly apply CRAWDAD to this type multi-cellular spot data, we would need to

know how many cells of each type are inside the spots and incorporate segmentation of the H&E images and cell-type deconvolution, which is outside the scope of this project. On the other hand, super resolution spot level methods, such as Visium HD, that collect information with subcellular resolution could be aggregated into cells for CRAWDAD analysis.

We have now added a clarification of this to the discussion, provided below for the reviewer's convenience:

Lines 394-399:

Finally, although we have elected to demonstrate CRAWDAD analysis on datasets from select SRO technologies, in general, CRAWDAD is amenable to any SRO technology for which spatial positions and associated labels can be derived. However, we caution that for some multi-cellular spot-based SRO technologies, additional deconvolution or processing may be needed to ensure appropriate interpretation of results. In general, we recommend applying CRAWDAD to datasets with single cell resolution to facilitate interpretation.

8. It would nice to see a comparison with the following approach: first using spatial grid to group cells and then calculating the correlation between cell proportion of two cell types in different distance scale (keep only grids that contain two cell types).

We thank the reviewer for this interesting suggestion. A similar approach has been explored previously in [Viladomat et al, Biometrika 2014] and was shown to lead to false positives due to the high-levels of spatial-autocorrelation in the features of interest (such as cell-type proportions). We can demonstrate this using our own data with the simulation described previously in Comment #3 and again assess type-I-error. As suggested, we created a spatial grid to group cells and then calculated the Pearson's correlation between the cell-type proportions of pairs of cell-types across all grids. We interpreted significant Pearson's correlations as significant co-localizations. Based on this approach, consistent with Viladomat et al, we were able to confirm that this approach gives a highly inflated type-I-error rate of approximately 43% consistent with the previously published findings.

9. For Figure 1c, why authors use large circle for 350 mum and smaller circles for 550 mum?

The reviewer is correct that we use a larger circle to correspond to a smaller scale and a smaller circle to correspond to a coarser scale. This is because we generally care more about co-localization relationships that reach significance at a smaller scale. Therefore, we use a bigger size to draw attention to the aspect of the data that we care more about. We have clarified this in the revised methods, provided below for the reviewer's convenience:

Lines 490-498:

To visualize the spatial relationship results for a particular cell-type pair across different spatial extents, CRAWDAD uses trend plots where the x-axis represents the length scale and the y-axis represents the Z-score such that the trend represents how the spatial relationship of the cell-type pair changes as the spatial length scales increases. To summarize relationship trends across all

evaluated cell-type pairs, CRAWDAD identifies the scale and value of the first Z-score of the trend that is above the significance threshold and plot them as a dot for each reference and neighbor pair and uses a dot plot where the Z-score value is represented by the color hue and the scale is represented by the size of the dot. Smaller scales are represented as larger dots to visually emphasize the potential importance of these small-scale colocalization relationships.

Reviewer #2

We thank the reviewer for their helpful feedback. To thoroughly address these points, we have opted to rewrite the manuscript as a full Article instead of a Brief Communication. We hope the longer length format will clarify these points for readers.

Major Comments

1. When it is first introduced in the manuscript, please provide a clear definition of "length scale" from both biological and computational perspectives.

We thank the reviewer for the comment and further explained the meaning of length scale. Length scale refers to the spatial extent in which we should analyze the tissue for the relationship to be statistically significant. In a biological perspective, it means the physical size of the tissue region we should look at to determine if the relationship between two cell types exists. In a computational perspective, it refers to the size of the squared grids that are used to create the null background.

We have now further clarified this in the introduction as well as updated Figure 1 to provide a visualize explanation, provided below for the reviewer's convenience:

Lines 49-58:

Cell-type spatial relationships can occur at different length scales, with some cell types colocalizing to engage in paracrine signaling and other close-range interactions at a fine, micrometer length scale³; others colocalizing into distinct environments and functional tissue units at a more meso-scale¹; while others colocalizing into anatomical structures at a more macro-scale (Fig 1a-b). Whether we consider two cell-types as being colocalized is often a function of the spatial extent that we analyze (Fig 1b). For example, two cell types uniquely present in distinct layers of the brain may be considered separated if we consider only the spatial extent of the brain. However, we may consider these cell types to be colocalized in the same organ if we consider the spatial extent of the whole body. Thus, we sought to consider the effects of spatial extent by investigating cell-type spatial relationships across different length scales.

We also added more information in the results section:

Lines 74-78:

Next, CRAWDAD creates a series of non-overlapping grids of tiles (square or hexagonal) where the side-length of each tile corresponds to a user-defined spatial length scale. Then, it shuffles the cell-type annotations for all cells within each tile to create an empirical null background at the specified spatial length scale (Fig 1d)

Figure 1. Motivating Cell-type Relationship Analysis Workflow Done Across Distances (CRAWDAD) using simulated data. **a**. Illustration of the cell-type spatial relationships found at different length scales. **b**. Simulated spatial omics tissue data, visualized at different scales. Each point is a cell, colored by cell type. Scale bars correspond to 250 μm . **c**. Representation of the creation of the neighborhood and the null background. CRAWDAD draws a circle (neighborhood distance as the radius) around each cell of the reference cell type and merges them into one

neighborhood. d. CRAWDAD creates a grid of side-by-side tiles (length scale defined as the side length for square pixels and the distance between opposite edges for hexagonal pixels) and shuffles the labels inside each tile to create the null background. e. The multi-scale spatial relationship trend plot for reference cell-type C and neighbor cell-type B. The horizontal dotted lines represent the z-score significance threshold corrected for multiple testing ($Z\text{-score} = \pm 2.96$). The red bars denote the standard error for the z-score estimated using permutations. f. The multi-scale spatial relationship trend plot for reference cell-type A and neighbor cell-type B. g. Summary visualization all cell-type spatial relationships. The size of the dot represents the scale in which a neighbor cell-type first reaches a significant spatial relationship with respect to a reference cell-type. The color of the dot is the z-score at such scale. Created in BioRender. Fan, J. (2023) BioRender.com/y47n964.

2. As 10x Visium is the predominant platform in the market, it is essential to demonstrate the method's compatibility with this data. Moreover, does the method align with the availability of various deconvolution (<https://doi.org/10.1038/s41467-023-40458-9>, <https://doi.org/10.1038/s41587-022-01273-7>) and super-resolution (<https://doi.org/10.1038/s41587-021-00935-2>, [10.1016/j.cels.2023.03.008](https://doi.org/10.1016/j.cels.2023.03.008)) methods?

We appreciate the reviewer's question. As CRAWDAD only considers spatial positions and their associated labels, technically, it could be applied to spot-based SRO. However, since spot-based SRO data may have spots with multiple cells, the results should be interpreted with care. We theorize that to properly apply CRAWDAD to this type of multi-cellular spot data, we would need to know how many cells of each type are inside the spots and incorporate segmentation of the H&E images and cell-type deconvolution, which is outside the scope of this project. On the other hand, super resolution spot level methods, such as Visium HD, that collect information with subcellular resolution could be aggregated into cells for CRAWDAD analysis.

We have now added a clarification of this to the discussion:

Lines 394-399:

Finally, although we have elected to demonstrate CRAWDAD analysis on datasets from select SRO technologies, in general, CRAWDAD is amenable to any SRO technology for which spatial positions and associated labels can be derived. However, we caution that for some multi-cellular spot-based SRO technologies, additional deconvolution or processing may be needed to ensure appropriate interpretation of results. In general, we recommend applying CRAWDAD to datasets with single cell resolution to facilitate interpretation.

3. The dataset for demonstration is limited to only three platforms, which is relatively small compared to the wide array of spatial platforms currently available. To achieve a wider scope for an interdisciplinary journal like Nat Comm, additional spatially resolved techniques, such as MIBI-TOF, nanostring-SMI, and MERSCOPE, etc., should be included in the analysis.

We thank the reviewer for this suggestion. We have now applied CRAWDAD to additional datasets from a wide array of spatial platforms including Xenium and MERSCOPE as suggested. In general, we emphasize that our approach is amenable to technologies for which we can obtain cell positions and cell-type labels for all cells in the tissue (as opposed to technologies that only characterize the spatial organization of one or two cell-types). We have added a clarification of this to the discussion. We hope this will help users understand how they may use CRAWDAD even as new spatial platforms become available:

Lines 217-222:

To further exemplify CRAWDAD's applicability to potentially less well-organized tissues such as cancer tissues, we applied it to a breast cancer dataset assayed by Xenium6 (Fig 3a). We applied CRAWDAD to evaluate 162,107 annotated cells representing 19 cell-types using a neighborhood size of 100 μ m across length scales ranging from 100 μ m to 1000 μ m (Fig 3, Methods). CRAWDAD identified three groups of cell types based on their cell-type spatial relationships, corresponding to histologically distinct structures (Fig 3b-e).

Fig 3. CRAWDAD characterizes cell type spatial relationships in breast cancer assayed by Xenium. a. Spatial visualization of annotated cell types (left) with corresponding histology image (right). b. Summary visualization all cell-type spatial relationships in the breast cancer data. Select groups of consistently colocalized cell-types is outlined by a unique color. c. Spatial visualization of the consistently colocalized cell-types. Plots for each tissue structure are outlined by the corresponding color in (b). Scale bars correspond to 250 μ m.

Lines 246-275:

Beyond characterizing cell-type spatial relationships within a single sample, such multi-scale characterization enabled by CRAWDAD can also be used to compare cell-type spatial relationships across samples spanning different conditions, such as health and disease, development, or replicates. To demonstrate this functionality, we applied CRAWDAD to nine mouse brain samples assayed by MERFISH comprised of three replicates from three distinct Bregma locations⁷ with cell-type annotations obtained previously using unified clustering¹⁸ (Fig 4a). We applied CRAWDAD to evaluate 734,693 annotated cells across all datasets representing

14 cell-types using a neighborhood size of 50 μ m across length scales ranging from 100 μ m to 1000 μ m (Methods). To compare multi-scale cell-type spatial relationships across samples, we calculated the signed area under the curve (AUC) for each Z-score trend for each cell-type pair. We then performed dimensionality reduction with principal component analysis (PCA) on all signed AUC values to find that cell-type spatial relationships of replicates from the same Bregma location are highly similar, as they are positioned closer together in PC space (Fig 4b). We likewise overall observed a smaller variance in the signed AUC values within replicates from the same Bregma location compared to across locations (Fig 4c). These results suggests that samples from the same Bregma location have cell-type spatial relationships that are more similar than those from different Bregma locations, as expected. Importantly, this similarity in cell-type spatial relationships is robust to tissue rotation and small local diffeomorphisms, as some of the brain tissue sections profiled are rotated with small tissue distortions compared to others. To investigate specific highly variable cell-type spatial relationships further, we visualized the spatial-relationship trends for the cell-type pair with the highest signed AUC variance across locations: GABAergic Estrogen-Receptive Neurons as reference and Excitatory Neurons as neighbor (Fig 4d). Despite its comparatively higher signed AUC variance across locations, samples from the same Bregma location still generally exhibited the same depletion trend whereas samples across Bregma locations varied (Fig 4d). Visual inspection of GABAergic Estrogen-Receptive Neurons and Excitatory Neurons also suggested high consistency in terms of spatial relationships within replicates from the same Bregma location compared to across locations (Fig 4e). As such, cell-type spatial relationship trends quantified by CRAWDAD can be used to compare across samples to confirm that cell-type spatial relationships in the mouse brain are generally highly consistent within replicates from the same Bregma location compared to across locations.

Figure 4. CRAWDAD enables comparison of spatial relationships across different tissue sections of the mouse brain assayed by MERFISH. a. Spatial visualization of annotated cell types in each sample. Scale bars correspond to 1000 μ m. b. Visualization of each sample in the principal component space with the first two principal components calculated using the standardized AUC

of each multi-scale cell-type spatial relationship trend. c. Variability of multi-scale cell-type spatial relationship trends calculated as the variance of the AUC values across replicates (top) and locations (bottom). d. The multi-scale cell-type spatial relationship trend plot of GABAergic Estrogen-Receptive (ER) Neurons as reference and Excitatory Neurons as neighbor for samples of replicates from the same location (top) and different locations (bottom). e. Spatial visualization of the GABAergic ER Neurons and Excitatory Neurons in replicates from the same location (left) and different locations (right).

We also described these datasets in the methods section:

Lines 578-586:

Analysis of the human breast cancer

We collected the 7520.95 μ m-by-5471.17 μ m Xenium breast cancer dataset (in situ sample 1, replicate 1) and with annotated cell types from the original publication⁶. Supervised cell type annotations were used. We filtered the original data by removing cells with less than 3 gene counts, obtaining 162107 cells with x-y coordinates and 20 cell-type annotations.

For CRAWDAD, a set of shuffled null distributions were created at length scales of 100 to a 1000 by intervals of 100 microns for 3 permutations. A neighbor distance of 50 microns was used to evaluate every pairwise combination of cell-types at each length scale. To identify significant trends with multiple testing correction, we used a Z-score threshold of 3.84.

Lines 588-597:

Analysis of the mouse brains

We obtained the nine MERFISH mouse brain datasets from the Vizgen Data Release V1.0. May 2021⁷ with cell types previously annotated through unified clustering¹⁸. We filtered the original data by removing cells with less than 3 gene counts and merging sub-cell types. The resulting number of cells and cell-type annotations by sample is provided below:

Location	Replicate	Number of Cells	Number of Cell Types	Tissue Size
1	1	78329	13	9036.87x 6326.88
	2	88884	13	8060.09x 9936.43
	3	84635	14	8504.56x 8249.1
2	1	83546	14	8883.69x 7113.93
	2	84171	14	8867.34x 9316.67
	3	85957	14	9147.76x 6980.63
3	1	70844	14	7058.34x 7829.3

	2	83461	14	8952.54x 6747.24
	3	74866	14	8952.54x 6747.24

For CRAWDAD, a set of shuffled null distributions were created at length scales of 100 to a 1000 by intervals of 100 microns for 3 permutations. A neighbor distance of 50 microns was used to evaluate every pairwise combination of cell-types at each length scale.

Lines 636-645:

Comparing across different samples

To compare different samples, we opted to use the area under the curve (AUC) of the Z-score trend to represent each relationship instead of the scale of when the relationship reaches significance as not all of them do. We represented each sample by the AUC values of each cell-type pair, creating an AUC high-dimensional space. By applying principal component analysis to this space, we used the first two components to visualize the samples. In this case, instances that are similar in the high-dimensional space should also be similar in the low-dimensional one. Additionally, we investigated the variance of AUC across samples by plotting the variance for each cell-type pair in a dot plot. Lastly, we visualized the relationship trends for the cell-type pair with highest AUC variance across conditions.

-
4. The manuscript primarily focuses on highly organised tissues. It is vital to demonstrate how the method can be applied to tissues with less organisation, such as tumours and other complex diseases, to assess its general applicability.

We thank the reviewer for this suggestion. We have now applied CRAWDAD to Xenium spatial transcriptomics data from breast tumor tissue section to demonstrate its general applicability, even to potentially less organized tissues such as tumors, as suggested. The response to Comment 3 details this application.

-
5. Comparing colocalisation summaries between different treatment/condition/perturbation scenarios holds greater significance than evaluating a single tissue in isolation. The manuscript should provide further elaboration on how the method can be adjusted to derive statistical power in these comparative scenarios.

We agree with the reviewer and acknowledge the importance of comparing different conditions. To further develop this topic, we've now included new analyses of nine different mouse brain samples, composed of three replicates for three distinct locations. We also developed functions to summarize the relationships found in the multiple samples by calculating their signed AUC of the cell-type spatial relationship trends. In our analysis, we found that samples from the same location of the brain share the most relationships with each other when comparing with sample from other locations, as expected. Response to Comment 3 details this application.

Further, to specifically compare across patients to identify potential patient-specific differences, we compared multiple human spleen samples. Given that these samples all come from non-diseased tissues as part of the Human BioMolecular Atlas Program, we observed that most spatial cell-type relationships presented high consistency with low variability across all samples. However, we were able to identify some patient-specific trends, though the sample sizes evaluated here limit our ability to make general significant conclusions. We've added information about this analysis in the results and discussion section, provided below for the reviewer's convenience:

Lines 298-313:

To determine whether such cell-type spatial colocalization relationships are consistent across tissue sections and individuals, we further repeated these analyses with 837,952 cells from five additional spleen samples both within and across individuals. To ensure all datasets were annotated in a uniform manner, we applied batch-correction²⁴ and used a linear discriminant analysis model to transfer cell-type annotations to these new datasets (Methods, Supp Fig 5). We then applied CRAWDAD to identify similar cell-type spatial relationships corresponding to the WP and RP compartments both within and across individuals (Fig 5d, Supp Fig 6a). Further analyzing the variance of the relationship trend's AUC values (Fig 5e), we noticed that most cell-type spatial relationship trends were highly consistent across patients and samples, reflecting the ordered patterning of the functional tissue regions (Fig 5e-f, Supp Fig 7a). Select cell-type spatial relationship trends had patient-specific relationships, exhibiting consistent trends within replicates from the same patient but varying across patients, suggestive of potential patient-specific variation (Fig 5g, Supp Fig 7b). Other cell-type spatial relationship trends varied even within replicates, suggestive of potential tissue sample-specific patterns (Fig 5h, Supp Fig 7c). In general, we anticipate assessing these variations in cell spatial relationships can give insight into inter- and intra-individual variation linked to donor and tissue-specific features.

Lines 340-346:

Additionally, we emphasize that such quantified cell-type spatial relationships trends can be used to compare across SRO datasets and demonstrate its application in identifying consistent spatial trends within mouse brain replicates that are distinct across Bregma locations. We further apply CRAWDAD to characterize cell-type spatial relationships to HuBMAP SRO datasets of the human spleen to identify generally consistent spatial trends reflective of the organization of the red and white pulp but also reproducible patient-specific variation, though the sample sizes evaluated here limit our ability to make general significant conclusions.

Figure 5. CRAWDAD characterize cell-type spatial relationships in the human spleen assayed by CODEX. a. Heatmap of marker protein expression for annotated cell types. b. UMAP reduced-dimensional visualization of annotated cell-types. c. Spatial visualization of annotated cell-types in one representative tissue section. Scale bars correspond to 250 μ m. d. Summary visualization of the multi-scale cell-type spatial relationship analysis for tissue sections PKHL and XXCD from patient HBM966.VNKN.965. Cell types consistently colocalized in the white and red bulk are highlighted with small and large squares respectively. e. Variability of multi-scale cell-type spatial relationship trends calculated as the variance of the AUC values across samples. The multi-scale cell-type spatial relationship trend plots are shown for select cell-type pairs exhibiting f. low variability across different samples, g. high variability across patients but low variability within replicates, and h. high variability across samples including within patients. i. Subset of CD4+ Memory T cells near Follicle B cells. The number of CD4+ Memory T cells (n) and the proportion of subsets (left) and spatial visualization of subsets (right) in tissue sections PKHL and XXCD from patient HBM966.VNKN.965. j. Proportion of CD4+ Memory T cells near Follicle B cells over all CD4+ Memory T cells in each sample.

Again, due to the limited sample size, we cannot draw statistically significant conclusions regarding patient-specific trends here. Additional validation would be needed to corroborate findings that is beyond the scope of the current manuscript. However, as new spatial omics technologies emerge, we anticipate that comparisons of a much larger number of samples will enable such significant conclusions. We anticipate the need of additional meta-analysis statistical tools for users to derive statistical power from multiple information derived from CRAWDAD's analysis. We have added to the discussion to emphasize provide further elaboration on how CRAWDAD can be applied in comparative scenarios as suggested by the reviewer:

Line 342-354:

We further apply CRAWDAD to characterize cell-type spatial relationships to HuBMAP SRO datasets of the human spleen to identify generally consistent spatial trends reflective of the organization of the red and white pulp but also reproducible patient-specific variation, though the sample sizes evaluated here limit our ability to make general significant conclusions. As atlasing efforts such as HuBMAP¹⁹, the Human Cell Atlas²⁸, and others continue to profile the spatial organization of cells within tissues, we anticipate identifying significant spatial variation across axes of interest will become more feasible in the future, though additional scalable, comparative meta-analysis tools to integrate statistics from many samples across multiple studies in a manner that is robust to batch effects may be then needed. We expect that the incorporation of quantitative spatial trend metrics such as those provided by CRAWDAD will be useful in such meta-analyses to ultimately facilitate in the identification and characterization of cell-type colocalization relationships in complex tissues to advance our understanding of the relationship between cell-type organization and tissue function.

-
6. The definition of "cell type" significantly influences the method's output (e.g., inaccurate cell type identifications and granularity determination, etc.). The method can benefit from considering: (1) the cell type continuum in the gene expression space; (2) cases where cell typing is not perfectly accurate. This issue becomes particularly pronounced in cases such as Figure 3b, where the cell type boundary in the feature space is unclear.

We thank the reviewer for this important comment. Indeed, our method relies on accurate cell-type annotations. Besides, the granularity of those cell-type annotations will highly depend on the biological question of interest. Therefore, we recommend the user to apply methods currently available to better cluster the cells and to clean up cell type annotations, such as Clusteval [Taskesen, 2020] and NbClust [Charrad et al, Stat Softw 2014].

We have added a discussion on these limitations, provided below for the reviewer's convenience:

Line 387-393:

Third, since CRAWDAD takes annotated cell-type as input, the quality of the results directly depends on the quality of the annotation. Misannotated cell types could shift the proportions of other cell types inside spatial neighborhoods to alter the spatial relationships identified by CRAWDAD. Thus, cell-type annotations may be evaluated for robustness and cleaned if needed prior to CRAWDAD analysis^{31,32}. Or alternatively, identified cell-type spatial relationships may be re-evaluated given multiple potential cell-type annotations to ensure the robustness of identified trends.

-
7. The simulation is overly simplified and can be improved by designing multiple simulation studies, considering (1) a continuous cell state along the gene expression space, (2) mixed cell types in space, (3) inaccurate cell type identification, (4) noise in gene expression (Gaussian, Poisson), and (5) uneven spatial distributions.

We thank the reviewer for this suggestion. To provide additional, more complex simulated data with mixtures of cell types in space and uneven spatial distributions, we created ten simulated datasets with self-enrichment patterns using a simulation approach previously described [Viladomat et al, Biometrika 2014]. Briefly, we used a Matern autocorrelation function to create two independent gaussian random fields. We binarize the gaussian random field to create two spatially separated cell-types. Due to the autocorrelation, each cell-type should be colocalized with itself and separated from the other cell type originated in the same gaussian field. Likewise, given the underlying independent gaussian random fields, cell-type pairs across the two independent instantiations should exhibit no spatial relationship. We repeated this process to generate ten datasets. In this manner, these new simulated datasets have distinct uneven spatial distributions and unique mixtures of cell types in space, presenting a substantial improvement to our previous simplified simulation.

We were able to use these simulated datasets to benchmark CRAWDAD against other tools. We evaluated the ability of CRAWDAD, Ripley's K and Squidpy in distinguishing enrichment and depletion between cell types. For this problem, we considered detecting the relationship (enrichment or depletion) as a positive result and not detecting it as a negative result. In this manner, we calculated the number of false positives for each tool. As the Ripley's K and Squidpy's co-occurrence methods do not present a statistical threshold to detect significant relationships, to perform a fair comparison between our method and the others, we determined the cell-type relationships based on a relative comparison of the Z-score trends. The neighbor cell type which Z-score trend had the highest area under the curve (AUC) value the was considered enriched

around the reference cell type, while the one with the lowest AUC value was considered depleted. Since there are only two true relationships for each cell type in our simulated datasets, this strategy is able to detect them even though it does not rely on a statistical threshold.

We've added the results for these simulations using this more complex simulation approach to the revised manuscript, with relevant excerpts provided below for the reviewer's convenience:

Lines 140-175:

To further benchmark and compare CRAWDAD's functionality, we simulated a variety of SRO datasets using a previously developed simulation framework¹¹ (Methods). Briefly, we simulated cells by sampling from a uniform distribution to create x-y positions. We split the cells into two groups, and, for each group, we associated a value to each cell using independent, autocorrelated Gaussian random fields (Supp Fig 2a). We binarized the values, splitting the cells into two cell-types based on the underlying simulated value (Supp Fig 2b). In this manner, we created a simulated dataset with four cell types (Supp Fig 2c) where we expect each cell-type to be enriched with itself due our use of spatially autocorrelated simulation values. Likewise, we can expect the two cell-types simulated from the same Gaussian random field to be spatially mutually exclusive and therefore identified to be separated. Additionally, we expect the cell-types from different random fields to exhibit no significant spatial relationship due to the Gaussian random fields being independent. We repeated this process to create a total of ten random simulated datasets.

We used these simulated datasets to benchmark and compare CRAWDAD with two other spatial relationship analysis methods that also consider spatial length scales, Ripley's K Cross¹² and Squidpy's co-occurrence implementation of the approach described in Tosti et al.¹³ Although all evaluated methods perform cell-type enrichment analysis across length scales, their definition of length scales differs. Briefly, Ripley's K Cross evaluates multiple length scales by increasing the neighborhood size while comparing the cell-type proportion in the neighborhood to the global proportion. On the other hand, Squidpy's co-occurrence implementation evaluates multiple length scales by increasing the size of an annulus neighborhood and calculating the conditional probability of the neighbor cell types given the reference cell type. In addition, as Ripley's K Cross and Squidpy's co-occurrence implementation do not present a threshold to determine statistical significance, for comparative purposes, we opted to assess each method's ability to distinguish between cell-type spatial enrichment and depletion. Specifically, given a reference cell type, we considered a method as achieving a true positive prediction if the cell type identified with the most enriched relationship trend was itself. Alternatively, we also considered a method as achieving a true positive prediction if the cell type identified with the most depleted relationship trend was the other cell type from the same Gaussian random field. We identified the cell-type with the most enriched and most depleted spatial trend using their area under the trend curve value (Methods). Using this approach, we evaluated all four cell types across all ten simulated datasets using all three methods (Supp Fig 2d). Based on this simulation framework, we obtained a true positive rate of 0.95 for CRAWDAD, 0.86 for Squidpy's co-occurrence implementation, and 0.8 for Ripley's K Cross. In this manner, cell-type spatial relationships identified by CRAWDAD can more accurately distinguish between cell-type spatial enrichment and depletion compared to other evaluated methods based on simulated data.

Supplementary Figure 2. Creation of the simulated dataset. a. Two independent Gaussian random fields with uniformly sampled spatial positions representing 1000 cells. b. These spatial positions are assigned a cell type based on the random field's value. c. Both fields are combined to generate one dataset with 4 cell-types. d. Relationships trends obtained by each method using cell-type A as the reference cell-type.

Additionally, we explained how the simulation was created in the methods:

Lines 536-545:

Simulating SRO data using Gaussian random fields

To create the simulated datasets with self-enrichment patterns, we followed the procedure previously described¹¹. First, we simulate the position of 2000 cells by sampling from a uniform distribution ranging from 0 to 1 for both x and y axis. Then, we randomly split the cells in two groups of 1000 instances each. Using the Matern function with nugget variance of 0.1, shape parameter of 0.5, and smoothness parameter of 0.3, we created a covariance function to generate a Gaussian random field for each group. We binarize each field by assigning positive cells to one cell type and negative cells to the other. Finally, we merge both groups and scaled the cells' positions to 1000 microns. To generate all the ten datasets, we repeated this process using a different random seed for each.

We also explained how the methods were compared in the methods:

Lines 547-553:

Comparing methods using simulated data

To benchmark CRAWDAD without relying on a significance threshold we performed a relative comparison between the relationship trends. We classified a cell-type as enriched in the neighborhood of the reference cell type if it had the trend with the highest area under the curve (AUC) value in the reference cell-type trend plot. Likewise, we classified a cell type as depleted if it had the most negative AUC value. We focused on measuring each method's capacity to distinguish trends, not the ability to identify statistically significant results.

8. We found the case study on Slide-seq data in Figure 2 unreliable. Slide-seq (V1/V2) is known to capture parts of multiple or partial cells for each spot, leading to unrealistic cell type and spatial localisation. It appears that the proposed method does not consider this.

The reviewer is correct that Slide-seq does not provide true single-cell resolution. As such, the results will need to be interpreted with this in consideration. We have now further clarified this in the results section:

Lines 182-188:

Because Slide-seqV2 uses 10 μ m barcoded beads to profile the gene expression within tissues in a spatially resolved manner, spatially resolved measurements may not necessarily correspond to single cells. However, given that a typical animal cell is also roughly 10-20 μ m in size¹⁵, we assumed here that the observed spatial position and cell-type assignments associated with each bead generally reflects the spatial position and cell-type annotations of the cell within the immediate vicinity of that bead. As such, we treat Slide-seqV2 beads with non-doublet RCTD annotations as effectively single cells for CRAWDAD analysis.

We have further added to the discussion a note on potential applications to spot-based Visium data, which has an even coarser resolution. Again, while CRAWDAD can be run on data from these technologies, it will be important to interpret the results in the context of the data. We provide the excerpt of the discussion in response to Comment #7.

9. Furthermore, for the Slide-seq case, it is valuable to see the proposed method's output align with known biology. However, for a method paper, there should be a quantitative comparison between the proposed method and others.

We thank the reviewer for the suggestion and added a quantitative comparison to the paper, as described in response to Comment #7. Additionally, we updated Fig 2 to highlight the trends calculated for each method given a specific cell type pair and extended the results section:

Lines 196-207:

Additionally, we applied Ripley's K Cross (Fig 2d) and Squidpy's co-occurrence implementation (Fig 2e) to the same dataset. We find that these other methods do not as clearly distinguish these

expected cell-type spatial relationships. Specifically, when analyzing cell-type spatial relationships with Purkinje neurons as the reference cell-type, we note that CRAWDAD's Z-score trend for Bergmann glia increases as the length scale increases, crossing the upper significance threshold and defining an enrichment of Bergmann glia among the neighborhood of Purkinje neurons as expected (Fig 2c). Likewise, CRAWDAD's Z-score trend for oligodendrocytes decreases as the length scale increases, crossing the lower significance threshold and defining a depletion of oligodendrocytes among the neighborhood of Purkinje neurons as expected (Fig 2c). These two cell-type trends are further distinct from other cell-types in the cerebellum. This clear separation between these two cell-type trends is not observed in the other evaluated spatial analysis methods (Fig 2d-e).

Lines 214-216:

Again, such differences between cell-type spatial relationships are difficult to discern using other evaluated spatial analysis methods (Fig 2i-j).

Figure 2. CRAWDAD characterizes cell-type spatial relationships in the mouse cerebellum assayed by Slide-seqV2 and the mouse embryo assayed by seq-FISH. a. Spatial visualization of cell type annotations from RCTD in the cerebellum. Scale bars correspond to 250 μ m. b. Summary visualization all cell-type spatial relationships in the cerebellum data. Select cell types highlighted to correspond with (c-e). c-e. The multi-scale spatial relationship trend plot for Purkinje neurons

as the reference cell type for (c) CRAWDAD, (d) Ripley's K Cross and (e) Squidpy co-occurrence implementation of Tosti et al. with neighboring cell-types Bergmann glia and Oligodendrocytes highlighted in red and blue respectively. All other neighboring cell-types in grey. f. Spatial visualization of annotated cell types in the embryo data. Scale bars correspond to 250 μ m. g. Summary visualization all cell-type spatial relationships in the embryo data. Select cell types highlighted to correspond with (h-j). h-j. The multi-scale cell-type spatial relationship trend plot for Endothelium cells as the reference cell type for (h) CRAWDAD, (i) Ripley's K Cross and (j) Squidpy co-occurrence implementation of Tosti et al. with neighboring cell-types Cardiomyocytes and Lateral plate mesoderm highlighted in red and blue respectively. All other neighboring cell-types in grey.

10. In Figure S2d-f, it is unclear why CRAWDAD is superior to Squidpy and Ripley based on the figure. The author should clearly define what the golden standard ground truth is for each dataset to determine which method is better. Similar issues arises in Figure S2g-I and Figure S2j-l. Please establish a more solid and comprehensive benchmarking analysis in the manuscript.

We appreciate the opportunity to clarify the difference between CRAWDAD, Ripley's K Cross, and the spatial co-occurrence method developed by Tosti et al and implemented in Squidpy. As explained in response to Comment #9, CRAWDAD allows a clearer distinction between the colocalization and separation of two cell types in the mouse cerebellum and mouse embryo. Based on the new simulation framework described in response to Comment 7 as suggested by the reviewer, CRAWDAD also achieves a superior true positive rate (0.95 for CRAWDAD, 0.86 for Squidpy's co-occurrence implementation, and 0.8 for Ripley's Cross K) in terms of distinguishing between cell-type colocalization relationships. We believe these comparisons using both real and simulated spatial datasets now provide a solid and comprehensive benchmarking analysis as suggested by the reviewer. We have moved the comparison figures from the supplement to the main figure 2 as addressed in Comment #9 above to better emphasize these differences. We also updated the description of each tool in the methods section for further clarification:

Lines 648-658:

Ripley's Cross-K Analysis

Ripley's Cross K function draws a circular neighborhood around each reference cell, counts the number of cells of each type inside this region, and divide it by the cell-type density. This value is compared to the theoretical K. The multi-scale aspect of this analysis comes from varying the neighborhood size. Additionally, cells in the border of the tissue will consider areas that do not present any cell, requiring the application of border correction methods to mitigate this effect.

We used the spatstat (version 3.0-6) package³⁸ to compute different Ripley's Cross-K values for each pairwise combination of cell types. To compare with the theoretical K and perform border correction, we subtracted the theoretical K for a Poisson homogeneous processes from the isotropic edge corrected Ripley's Cross-K. For consistency in visualization, we set the maximum radius size to be the same as the maximum length scale evaluated in CRAWDAD.

Lines 666-673:

Squidpy's Co-occurrence Probability

Squidpy¹⁰ implements the co-occurrence probability method originally presented in Tosti et al.¹³ The function works by drawing annular neighborhoods around each cell of the reference cell type. Then, it calculates the probability of a cell type being enriched in that region. The multi-scale aspect of this analysis comes from varying the neighborhood size.

Using Squidpy (version 1.2.3) and its `co_occurrence` function, we calculated the co-occurrence probability of clusters for each cell type. For consistency in visualization, we set the maximum distance to be the same as the maximum length scale evaluated in CRAWDAD.

However, we emphasize that these tools do perform different spatial quantifications. Briefly, Ripley's K Cross creates circular neighborhood of increasing sizes around the reference cell type and compares the cell-type proportions calculated in each of them to the proportion in the whole tissue, providing only a global analysis. Of note, Ripley's K may generate artifacts once the neighborhood size considers spaces outside the tissue area. In Rev Fig 1a, we illustrate how these neighborhoods are created for one of the cells and how they can encompass areas outside the sample. Squidpy co-occurrence focuses on identifying cell types that are more probable to be together when investigating annulus regions around the reference cell. In Squidpy's default co-occurrence application, it first calculates the maximum and minimum distance between cells and divide their difference into regular intervals (Rev Fig 1b). Then, for each interval, it will create an annulus neighborhood around cells of the reference cell type, as illustrated for one cell in Rev Fig 1c. Lastly, it will calculate probability of the neighbor cell type conditional on the cell-type over the expected probability for that neighborhood size. In contrast, CRAWDAD identifies spatial relationships at multiple scales by shuffling cells using grids of different sizes. We hope this helps clarify the difference between these methods. As such, we anticipate which tool is "superior" may vary depending on the intended task. As we have demonstrated, CRAWDAD offers improved performance in terms of distinguishing between spatially colocalized and separated cell-types.

Reviewer Figure 1. Representation of the workflow of Ripley's K Cross and the Squidpy's implementation of the Toti et al co-occurrence function. **a.** Circular neighborhoods drawn around two of the cells of the reference cell type in the Ripley's K Cross analysis. **b.** Calculation of the minimum and maximum distances between cells to create the neighborhood intervals in Squidpy's co-occurrence implementation. **c.** Two annular neighborhoods drawn around one cell of the reference cell type in Squidpy's co-occurrence implementation.

11. A comparison of running time and memory usage against other related tools is necessary.

We thank the reviewer for the suggestion. However, in CRAWDAD, runtime and memory usage are greatly affected by user defined parameters such as the number of permutations, the number of scales, number of cores, etc. Therefore, as runtime and memory usage of our tool will largely depend on these user choices, we do not believe a comparison will provide a useful reference for readers.

Instead, we have updated our CRAWDAD software website to include runtime estimates for example analyses using real spatial omics datasets such as those included in this manuscript (ex. https://jef.works/CRAWDAD/2_seqfish). We hope this will provide users with a more realistic sense of possible runtimes. A screenshot of the CRAWDAD software website is provided below for the reviewer's reference:

Run pairwise analysis

```
## find trends, passing background as parameter
results <- crawdad::findTrends(seq,
                               dist = 100,
                               shuffle.list = shuffle.list,
                               ncores = ncores,
                               verbose = TRUE,
                               returnMeans = FALSE)
```

```
## Evaluating significance for each cell type

## using neighbor distance of 100

## Calculating for pairwise combinations

## Allantois

## Anterior somitic tissues

## Cardiomyocytes

## Cranial mesoderm

## Definitive endoderm

## Dermomyotome

## Endothelium

## Erythroid

## Forebrain/Midbrain/Hindbrain

## Gut tube

## Haematoendothelial progenitors

## Intermediate mesoderm

## Lateral plate mesoderm

## Low quality

## Mixed mesenchymal mesoderm

## Neural crest

## NMP

## Presomitic mesoderm

## Sclerotome

## Spinal cord

## Splanchnic mesoderm

## Surface ectoderm

## Time was 1.42 mins
```

```
## note: 1.73 minutes with 7 M2 cores
```

12. I am not sure if this be necessary element for a Nat Comm paper. Unique biological insights brought by the proposed method are lacking. The author should demonstrate how, by using CRAWDAD, individuals can identify previously unknown biological discoveries that cannot be identified by similar tools (such as Squidpy).

As explained in the response to Comment #10, CRAWDAD essentially performs a different analysis than Ripley's K Cross and the co-occurrence method. Therefore, the results presented by CRAWDAD are unique and not obtained with the other tools. As we have now demonstrated in Fig 2, certain cell-type spatial relationships are only discernable through CRAWDAD analysis. Likewise, unlike other methods, CRAWDAD can be used to compare spatial relationships across different samples and identify relationships that change across conditions, as seen in new Figs 4-5.

In general, the application of CRAWDAD to identify unique biological insights that are previously unknown is currently an active area of research in the lab. We emphasize that these insights will demand additional validation through orthogonal experimental follow-up prior to dissemination. We will look forward to sharing these results as a part of future publications with the broader scientific community.

Minor Comments

1. Consider splitting complex sentences, such as Line 44-48, into multiple sentences for improved clarity.

We thank the reviewer for the suggestion and have revised complex sentences accordingly.